# Chemogenetic stimulation of phrenic motor output and diaphragm activity

**Ethan S Benevides**[1,2,3], **Prajwal P Thakre**[1,2,3], **Sabhya Rana**[1,2,3], **Michael D Sunshine**[1,2,3], **Victoria N Jensen**[1,2,3], **Karim Oweiss**[3,4], **David D Fuller**[1,2,3]*

[1]Department of Physical Therapy, University of Florida, Gainesville, United States; [2]Breathing Research and Therapeutics Center, University of Florida, Gainesville, United States; [3]McKnight Brain Institute, University of Florida, Gainesville, United States; [4]Department of Electrical and Computer Engineering, University of Florida, Gainesville, United States

## eLife Assessment

The authors report that chemogenetic methods targeting the ventral cervical spinal cord can be used to increase phrenic inspiratory motor output and subsequent diaphragm EMG activity and ventilation in rodents. These findings are **important** because they are a necessary first step towards using chemogenetic methods to drive inspiratory activity in disorders in which motor neurons are compromised, such as spinal injury and degenerative disease. The data are **convincing**, with rigorous assessments of phrenic inspiratory activity and its ability to drive the diaphragm and subsequent ventilation, as well as assessments of DREADD expression.

*For correspondence:
dfuller@phhp.ufl.edu

**Competing interest:** The authors declare that no competing interests exist.

**Abstract** Impaired respiratory motor output contributes to morbidity and mortality in many neurodegenerative diseases and neurologic injuries. We investigated if expressing designer receptors exclusively activated by designer drugs (DREADDs) in the mid-cervical spinal cord could effectively stimulate phrenic motor output to increase diaphragm activation. Two primary questions were addressed: (1) does effective DREADD-mediated diaphragm activation require focal expression in phrenic motoneurons (vs. non-specific mid-cervical expression), and (2) can this method produce a sustained increase in inspiratory tidal volume? Wild-type (C57Bl/6) and ChAT-Cre mice received bilateral intraspinal (C4) injections of an adeno-associated virus (AAV) encoding the hM3D(Gq) excitatory DREADD. In wild-type mice, this produced non-specific DREADD expression throughout the mid-cervical ventral horn. In ChAT-Cre mice, a Cre-dependent viral construct was used to drive neuronal DREADD expression in the C4 ventral horns, targeting phrenic motoneurons. Diaphragm electromyograms (EMG) were recorded in isoflurane-anesthetized spontaneously breathing mice at 4–9 weeks post-AAV delivery. The DREADD ligand JHU37160 (J60) caused a bilateral, sustained (>1 hr) increase in inspiratory EMG bursting in both groups; the relative increase was greater in ChAT-Cre mice. Additional experiments in ChAT-Cre rats were conducted to determine if spinal DREADD activation could increase inspiratory tidal volume during spontaneous breathing, assessed using whole-body plethysmography without anesthesia. Three to four months after intraspinal (C4) injection of AAV driving Cre-dependent hM3D(Gq) expression, intravenous J60 resulted in a sustained (>30 min) increase in tidal volume. Subsequently, phrenic nerve recordings performed under urethane anesthesia confirmed that J60 evoked a >200% increase in inspiratory output. We conclude that targeting mid-cervical spinal DREADD expression to the phrenic motoneuron pool enables ligand-induced, sustained increases in phrenic motor output and tidal volume. Further development of this technology may enable application to clinical conditions associated with impaired diaphragm activation and hypoventilation.

## Introduction

Many respiratory disorders are associated with reduced or impaired activation of respiratory motoneurons. Neurologic injuries (e.g., traumatic spinal cord injury, stroke) and neurodegenerative conditions (e.g., amyotrophic lateral sclerosis, Pompe disease) will often result in decreased respiratory motor output, including impaired activation of the phrenic motoneurons which innervate the diaphragm (*Berlowitz et al., 2016*; *Brown et al., 2006*; *Perrin et al., 2004*; *Mehta, 2006*; *Burakgazi and Höke, 2010*; *Fuller et al., 2013*). Another prominent example is obstructive sleep apnea, in which pharyngeal motoneurons have reduced output during sleep (*Jordan and White, 2008*). Treatment options that increase respiratory motoneuron activation to improve breathing are limited. However, designer receptors exclusively activated by designer drugs (DREADDs) may have use in this regard (*Doyle et al., 2021*). Structurally derived from naturally occurring G-protein-coupled receptors, DREADDs have been engineered to respond exclusively to exogenous ligands that are otherwise biologically inert (*Zhu and Roth, 2014*; *Alexander et al., 2009*; *Armbruster et al., 2007*). This provides a means to selectively stimulate cells expressing the DREADD. Prior studies have used DREADDs to stimulate upper airway muscle activation during breathing (*Doyle et al., 2021*). For example, following expression of DREADDs in murine hypoglossal motoneurons, tongue electromyogram (EMG) activity can be increased using DREADD ligands (*Fleury Curado et al., 2017*; *Fleury Curado et al., 2018*; *Fleury Curado et al., 2021*; *Singer et al., 2022*; *Horton et al., 2017*). This response is functionally beneficial as shown by increased patency of the upper airway (*Fleury Curado et al., 2017*).

The present study focused on chemogenetic activation of the phrenic neuromuscular system. Phrenic motoneurons provide motor innervation of the diaphragm muscle and are located in the mid-cervical (C3–C5) spinal cord (*Fuller et al., 2022*). We tested the hypothesis that expressing DREADDs in the mid-cervical spinal cord would enable systemic (intravenous [IV] or intraperitoneal) delivery of a selective DREADD ligand to produce sustained increases in the respiratory-related activation of the diaphragm. In doing so, we addressed two important questions. First, we determined if effective diaphragm activation requires focal DREADD expression targeting phrenic motoneurons only, or if non-specific expression in mid-cervical interneurons and phrenic motoneurons would be sufficient. This question derives from prior studies of cervical spinal cord stimulation. A compelling body of work, with studies in multiple species, demonstrates that non-specific activation of cervical spinal networks can be highly effective at increasing diaphragm activation (*DiMarco and Kowalski, 2013*; *Jensen et al., 2019b*; *Alilain et al., 2008*). One theory to explain this result is that a general increase in the excitability of cervical propriospinal networks leads to increased phrenic motoneuron activation (*Satkunendrarajah et al., 2018*; *Jensen et al., 2019a*; *Lane, 2011*). There is also evidence that phrenic motoneurons can integrate multiple synaptic inputs in a manner that produces orderly recruitment (*DiMarco and Kowalski, 2013*). Accordingly, DREADD-induced activation of mid-cervical neurons or networks may be sufficient for ligand-induced diaphragm activation. On the other hand, DREADD expression may need to be restricted to phrenic motoneurons if the goal is to produce inspiratory-related diaphragm activation. To address this question, we studied diaphragm responses in two mouse models: (1) a wild-type model in which DREADDs were non-specifically expressed in the C3–C5 spinal cord, encompassing interneurons populations and motoneurons, and (2) a choline acetyltransferase (ChAT)-Cre model in which DREADD expression was restricted to ChAT-positive neurons in the ventral C3–C5 spinal cord, targeting the phrenic motoneuron pool.

The second question we addressed was if phrenic motoneuron activation via cervical spinal cord DREADDs could produce sustained increases in inspiratory tidal volume in unanesthetized, spontaneously breathing animals. While previous results from the hypoglossal motor system (*Fleury Curado et al., 2017*; *Fleury Curado et al., 2021*; *Singer et al., 2022*; *Horton et al., 2017*) provide a proof-of-concept that DREADDs can stimulate respiratory motoneuron activity, whether a sustained increase in tidal volume could be evoked by expressing DREADDs in phrenic motoneuron was uncertain. For example, during spontaneous breathing, a DREADD-induced increase in phrenic motoneuron excitability, and thus diaphragm activation, could be rapidly offset by decreases in bulbospinal neural drive to the phrenic motor pool, secondary to reduced arterial $CO_2$ or increased vagal-mediated inhibition. An increase in diaphragm activation could also trigger a decrease in accessory respiratory muscle activation, thereby attenuating or preventing increases in tidal volume. Lastly, data from the hypoglossal motor system (*Fleury Curado et al., 2017*; *Fleury Curado et al., 2021*; *Singer et al., 2022*; *Horton et al., 2017*), as well as our initial results in the anesthetized mouse indicated that

both phasic (i.e., during the inspiratory period) and tonic (i.e., occurring across the respiratory cycle) activation of the diaphragm would increase after DREADD activation, and how this would impact tidal volume was not clear. Accordingly, we studied ChAT-Cre rats using whole-body plethysmography and a direct measure of phrenic motor output via nerve recordings. The plethysmography studies allowed us to determine if DREADD activation of phrenic motoneurons causes a sustained increase in tidal volume and ventilation during spontaneous breathing in the unanesthetized rat. Phrenic nerve recordings were performed under urethane anesthesia and enabled direct quantification of DREADD activation on the neural drive of the diaphragm while controlling variables including arterial $CO_2$ and lung volume.

Collectively, the results of this work demonstrate that mid-cervical spinal DREADD expression enables the selective DREADD ligand, J60, to produce sustained increases in the neural drive to the diaphragm, resulting in increased tidal volume during spontaneous breathing. Further development of this technology may enable application to clinical conditions associated with impaired diaphragm activation and hypoventilation.

## Results
### Diaphragm EMG responses in wild-type mice
Wild-type mice underwent bilateral injections of AAV9-hSyn-HA-hM3D(Gq)-mCherry into the ventral horns at spinal segment C4. Following a 4- to 9-week incubation period, mice underwent terminal diaphragm EMG recordings before and after application of the selective DREADD ligand, J60. On average, wild-type mice showed increases in diaphragm EMG output in at least one hemidiaphragm after J60 administration (*Figure 1*). The area under the curve (AUC) of the rectified and integrated diaphragm EMG was significantly increased after DREADD activation (*Figure 1d*) in both the left (p = 0.002; *Supplementary file 1*) and right (p = 0.002; *Supplementary file 1*) hemidiaphragm. Additionally, the peak-to-peak amplitude of the rectified and integrated diaphragm EMG burst activity increased bilaterally following J60 administration (left hemidiaphragm: p = 0.056; right hemidiaphragm: p = 0.01; *Figure 1e*; *Supplementary file 1*). Lastly, the tonic activity of the diaphragm was assessed (*Figure 1f*). Similar to the previous measures of EMG output both the left (p = 0.052; *Supplementary file 1*) and right (p < 0.001; *Supplementary file 1*) hemidiaphragm exhibited an increase in tonic activity following J60 administration. Respiratory rate was consistent for the duration of the experiment. Notably, there was no substantial change in the respiratory rate of these spontaneous breathing mice after J60 administration (p = 0.863; *Figure 1g*; *Supplementary file 1*).

In all experiments, the J60 ligand produced an increase in diaphragm EMG burst amplitude during inspiration. However, this increase was not always detected in both the left and right hemidiaphragm EMG recordings. Five mice showed a bilateral increase in diaphragm output after J60 administration (*Figure 1*), four mice showed a response that was limited to the right hemidiaphragm, and two showed a response that was limited to the left hemidiaphragm (*Figure 1*).

### Diaphragm EMG responses in ChAT-Cre mice
ChAT-Cre mice received bilateral intraspinal injections of AAV9-hSyn-DIO-hM3D(Gq)-mCherry into the ventral horns at C4. ChAT-Cre mice underwent terminal diaphragm EMG recording using the same protocol as wild-type mice, following a 4- to 9-week incubation. All mice (*n* = 9/9) showed an increase in diaphragm EMG output in response to the J60 DREADD ligand in at least one hemidiaphragm (*Figure 2*). On average, both left (p = 0.011; *Supplementary file 2*) and right (p < 0.001; *Supplementary file 2*) diaphragm EMG AUC increased over time after J60 administration (*Figure 2d*). Diaphragm EMG peak-to-peak amplitude had a similar, bilateral increase following J60 delivery (left hemidiaphragm: p = 0.013; right hemidiaphragm: p < 0.001; *Figure 2e*; *Supplementary file 2*). Lastly, tonic activity also showed an increase over time after J60 administration (*Figure 2f*) for both the left (p = 0.002; *Supplementary file 2*) and right (p < 0.001; *Supplementary file 2*) hemidiaphragm. Respiratory rate decreased significantly over time after J60 administration (p < 0.001; *Figure 2g*; *Supplementary file 2*). Apart from two mice that showed a unilateral EMG response that was limited to the right hemidiaphragm the remaining ChAT-Cre mice (*n* = 7/9) had bilateral increases in diaphragm EMG output following J60 administration (*Figure 2*).

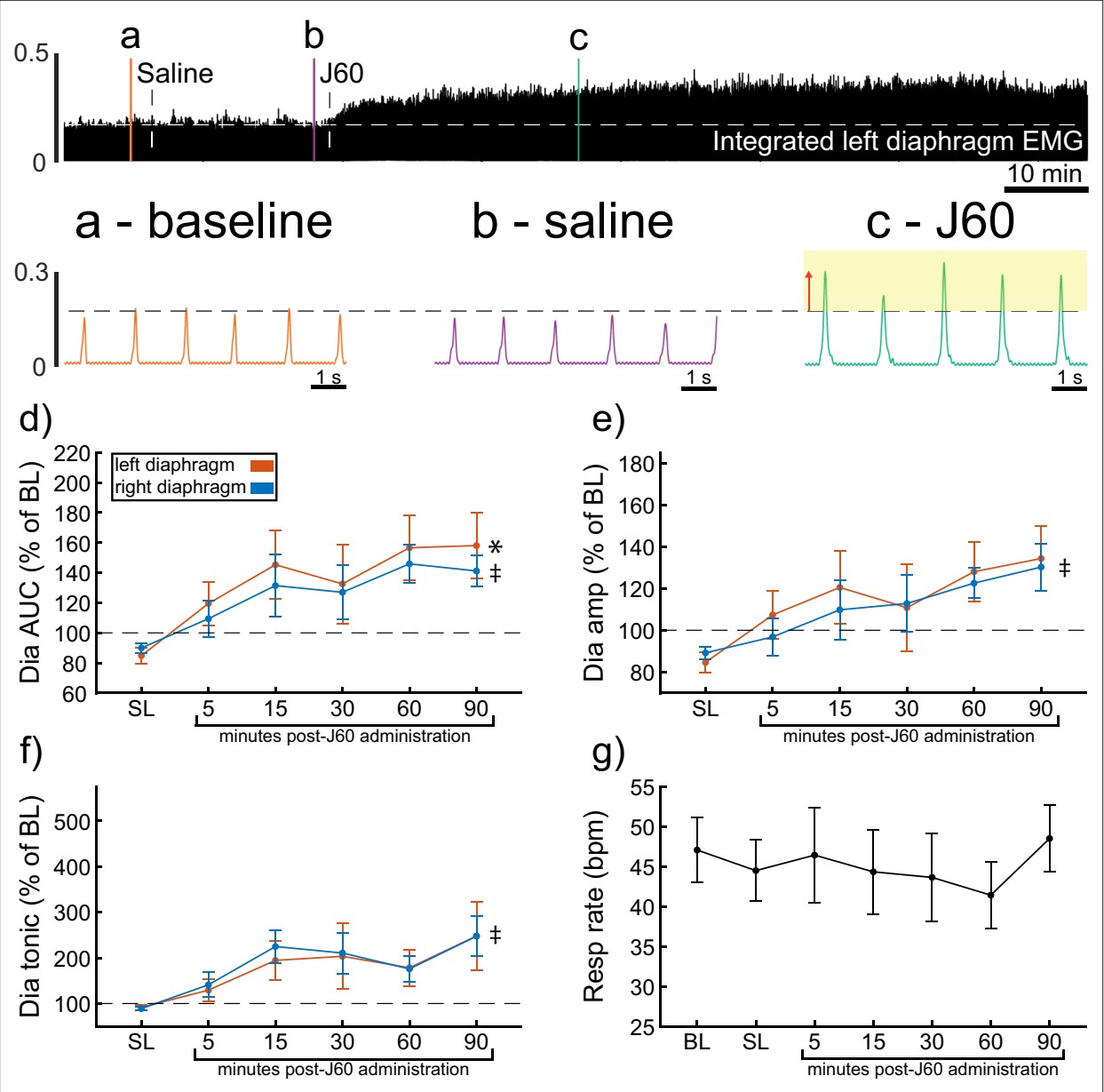

**Figure 1.** DREADD activation increases diaphragm electromyogram (EMG) output in wild-type mice. A representative example of diaphragm EMG activity before and after application of the J60 DREADD ligand is shown in the top panel. Examples of the individual inspiratory EMG bursts at baseline (**a**), after vehicle (**b**), and after J60 (**c**) are shown in call out panels below. The J60 ligand increased diaphragm output but did not impact respiratory rate. The mean responses (*n* = 11; *n* = 7 females) for EMG AUC, peak-to-peak amplitude, tonic activity, and respiratory rate are shown in panels d–g. For diaphragm EMG data (panels d–f), left hemidiaphragm EMG is represented in orange, while right hemidiaphragm EMG is blue. Error bars depict ±1 SEM. Statistical reports for all panels are provided in **Supplementary file 1**. * and ‡ symbols indicate significant main effects (p < 0.05) on one-way RM ANOVA for the left and right hemidiaphragm, respectively. Dia = diaphragm, AUC = area under the curve, amp = peak amplitude, BL = baseline, SL = saline (sham injection).

## Wild-type versus ChAT-Cre comparison

Diaphragm EMG responses of wild-type and ChAT-Cre mice were compared at the 30-min post-J60 time point (*Figure 3*). Left hemidiaphragm responses to J60 were similar between the two groups across all outcome measures (AUC: p = 0.998; peak-to-peak amplitude: p = 0.771; tonic activity: p = 0.160; *Figure 3a–c*; *Supplementary file 3*). However, right hemidiaphragm responses to J60 differed across AUC (*Figure 3d*), peak-to-peak amplitude (*Figure 3e*), and tonic activity (*Figure 3f*) with ChAT-Cre mice on average having larger magnitude responses compared to wild-type mice (AUC: p

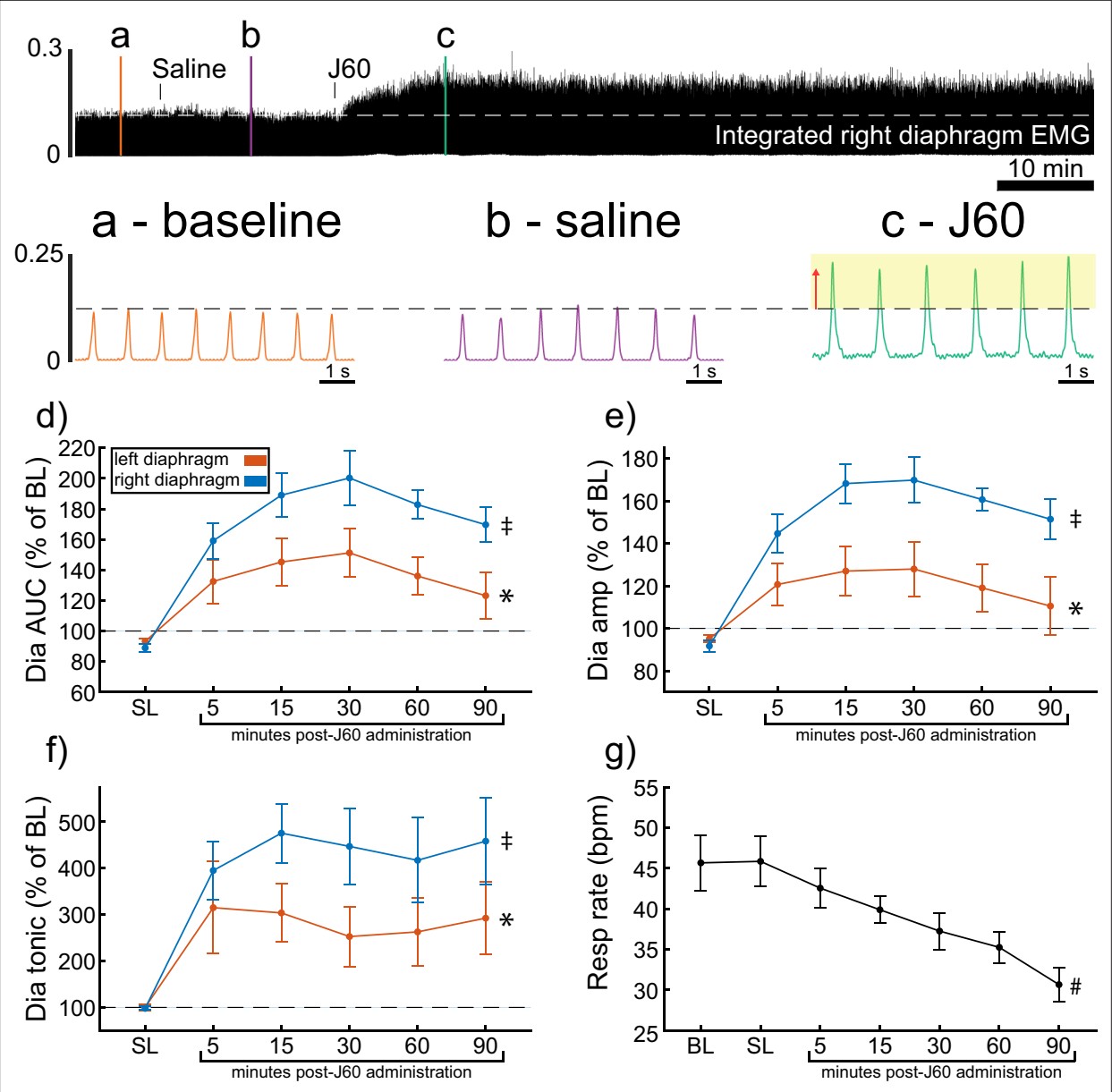

**Figure 2.** DREADD activation increases diaphragm electromyogram (EMG) output in ChAT-Cre mice. A representative example of diaphragm EMG activity before and after application of the J60 DREADD ligand is shown in the top panel. Examples of the individual inspiratory EMG bursts at baseline (**a**), after vehicle (**b**), and after J60 (**c**) are shown in the call out panels below. Mean responses ($n = 9$; $n = 6$ females) for EMG AUC, peak-to-peak amplitude, tonic activity, and respiratory rate are shown in panels d–g. The DREADD ligand caused a bilateral increase in diaphragm EMG AUC, peak-to-peak amplitude, and tonic activity. For all EMG parameters, the responses were greater on the right versus left hemidiaphragm. Respiratory rate decreased over time. For panels d–f, the left hemidiaphragm EMG is represented in orange, while right hemidiaphragm EMG is blue. Error bars depict ±1 SEM. Statistical reports for all panels are provided in ***Supplementary file 2***. * and ‡ symbols indicate significant main effects ($p < 0.05$) on one-way RM ANOVA for the left and right hemidiaphragm, respectively. # indicates a significant main effect ($p < 0.05$) on one-way RM ANOVA for respiratory rate data. Dia = diaphragm, AUC = area under the curve, amp = peak amplitude, BL = baseline, SL = saline (sham injection).

The online version of this article includes the following figure supplement(s) for figure 2:

**Figure supplement 1.** mCherry expression in the mid-cervical ventral horn corresponds with the laterality of the diaphragm electromyogram (EMG) DREADD response.

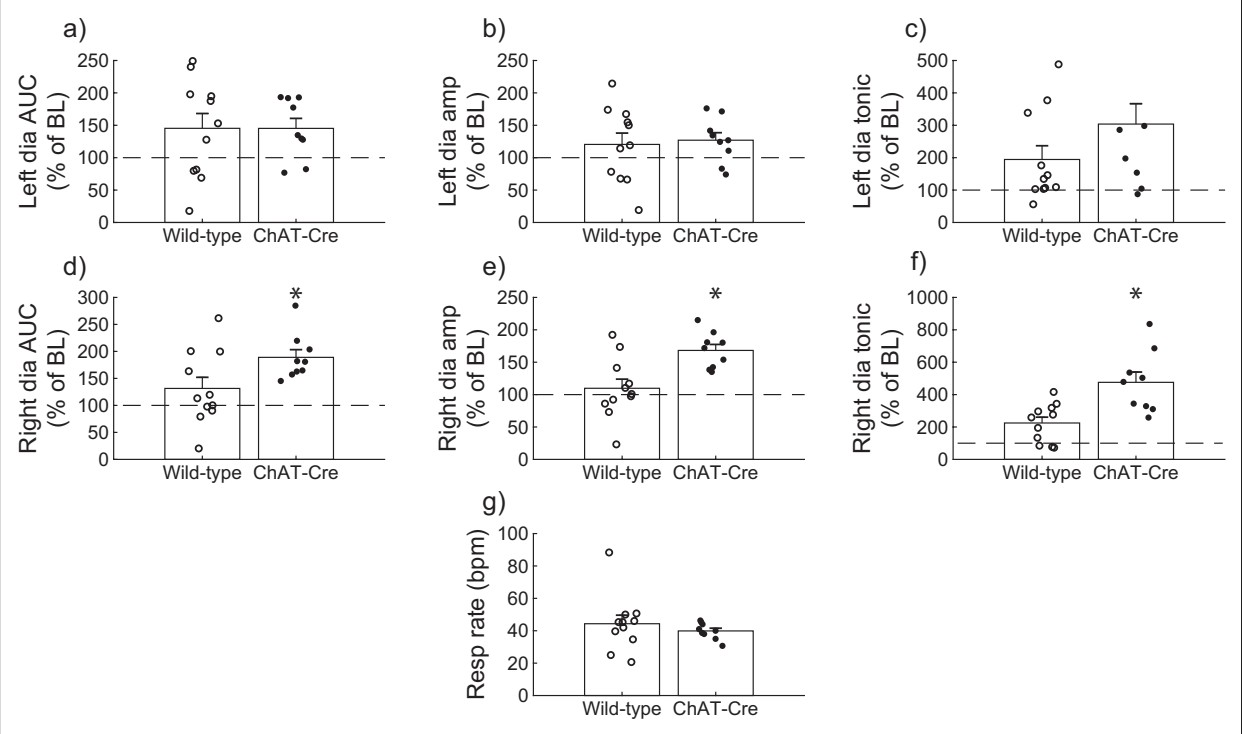

**Figure 3.** Wild-type versus ChAT-Cre mouse responses to DREADD activation. Direct comparisons of diaphragm electromyogram (EMG) response parameters (**a–f**) and respiratory rate (**g**) at 30-min post-J60 application (wild-type, $n = 11$; $n = 7$ females; ChAT-Cre, $n = 9$; $n = 6$ females). Left hemidiaphragm EMG AUC (**a**), peak-to-peak amplitude (**b**), and tonic activity (**c**) were similar between groups. However, the same parameters on the right hemidiaphragm (**d–f**) were greater in ChAT-Cre mice. Respiratory rate was similar between groups. Error bars depict ±1 SEM. Statistical reports for all panels are provided in *Supplementary file 3*. * indicates significant p value ($p < 0.05$) on paired t-test. AUC = area under the curve, amp = peak EMG amplitude, Dia = diaphragm, BL = baseline, resp rate = respiratory rate.

= 0.0417; peak-to-peak amplitude: p = 0.00403; tonic activity: p = 0.00207; *Supplementary file 3*). Respiratory rate was not different between the two groups (p = 0.382; *Figure 3g*; *Supplementary file 3*).

The a priori expected recording duration for these experiments in anesthetized and spontaneously breathing mice was 90 min. However, five of the eleven total wild-type mice in this experiment did not survive for this duration. It is unclear if this was a non-specific result associated with prolonged anesthesia, or if this was physiologically related to DREADD activation. No mice had evidence of adverse reaction in the initial 30 min following delivery of J60. Of the five mice which did not survive, $n = 3$ mice died between the 30- and 60-min time points after J60, and $n = 2$ mice died just prior to the 90-min time point. In contrast, all mice in the ChAT-Cre cohort ($n = 9/9$) survived the total duration of the experimental protocol. A Chi-square evaluation of the survival proportions was not statistically significant (Chi-squared = 3.2997, df = 1, p = 0.06929). However, considering the sample size, the results suggest some association between mouse strain (i.e., wild-type, ChAT-Cre) and death, suggesting that non-specific DREADD activation may be contraindicated.

## J60 control experiments

The DREADD ligand was administered to wild-type animals with no hM3D(Gq) expression in the mid-cervical spinal cord (*Figure 4*). This was done to determine the impact of J60 administration on diaphragm EMG in the absence of DREADD expression ($n = 2$ C57/bl mice; $n = 3$ Sprague Dawley rats). There was no discernable impact of J60 on the diaphragm EMG burst amplitude (mV) (left hemidiaphragm: p = 0.829; right hemidiaphragm: p = 0.496; *Figure 4a, b*). Responses were also not different between sham (saline) and J60 when normalized to baseline activity (left hemidiaphragm: p = 0.832; right hemidiaphragm: p = 0.376; *Figure 4c, d*).

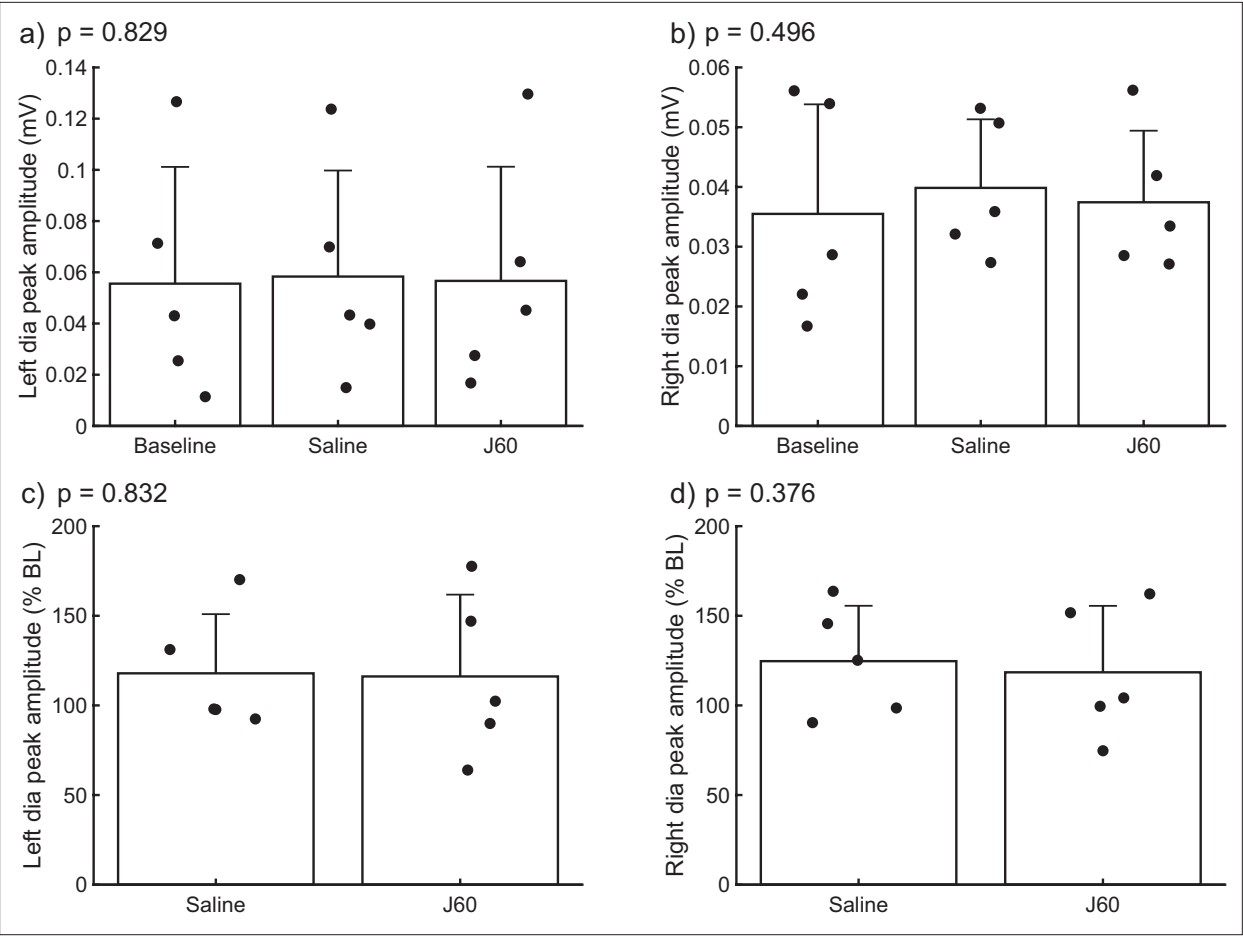

**Figure 4.** Impact of J60 application on diaphragm electromyogram (EMG) in the absence of hM3D(Gq) expression. Summary plots of a combined mouse ($n = 2$) and rat ($n = 3$) data showing the impact of J60 application (0.1 mg/kg) on diaphragm EMG peak amplitude in animals not expressing the hM3D(Gq) DREADD. Mean diaphragm EMG responses during baseline, saline (sham injection), and following J60 administration for the left hemidiaphragm are shown in panels a and b, respectively. Mean values normalized to baseline are shown in panels c (left hemidiaphragm) and d (right hemidiaphragm). Raw values (**a, b**) were assessed by one-way RM ANOVA while baseline normalized values (**c, d**) were assessed via paired *t*-tests. No statistically significant differences ($p < 0.05$) were detected in either hemidiaphragm across experimental periods. p-values are displayed next to each panel legend. Error bars depict ±1 SEM.

## ChAT-Cre rats – plethysmography and phrenic nerve recordings

A small cohort of ChAT-Cre rats underwent anesthetized diaphragm EMG recordings to ensure DREADD responses similar to the mouse cohorts could be obtained in rats. ChAT-Cre rats ($n = 4$) underwent bilateral, intraspinal injections of AAV9-hSyn-DIO-hM3D(Gq)-mCherry into the ventral horns at C4 to introduce the hM3D(Gq) DREADD transgene into phrenic motoneurons. Four of four rats showed increased diaphragm EMG output after DREADD activation (*Figure 6—figure supplement 1*). With that knowledge, we used a separate group of ChAT-Cre rats ($n = 9$; $n = 3$ females) to assess the effects of DREADD activation on ventilation. Whole-body plethysmography was used to measure breathing frequency, tidal volume, and minute ventilation before and after IV delivery of saline (sham) and J60 (*Figure 5*). Delivery of the J60 ligand resulted in an increase in inspiratory tidal volume compared to sham infusion (Normalized to Weight (ml/kg): Main effect of Treatment: $p = 0.037$; *Figure 5a*; Normalized to Baseline: Main effect of Treatment: $p = 0.091$; *Figure 5d*; *Supplementary file 4*). Respiratory rate appeared to be unaffected by DREADD activation and was similar between sham and J60 conditions (Respiratory Rate: Main effect of Treatment: $p = 0.582$; *Figure 5b*; Respiratory Rate Normalized to Baseline: Main effect of Treatment: $p = 0.774$; *Figure 5e*; *Supplementary file 4*). Minute ventilation was slightly elevated in the J60 condition versus sham; however, this increase did not reach the threshold for statistical significance (Normalized to Body Weight: Main

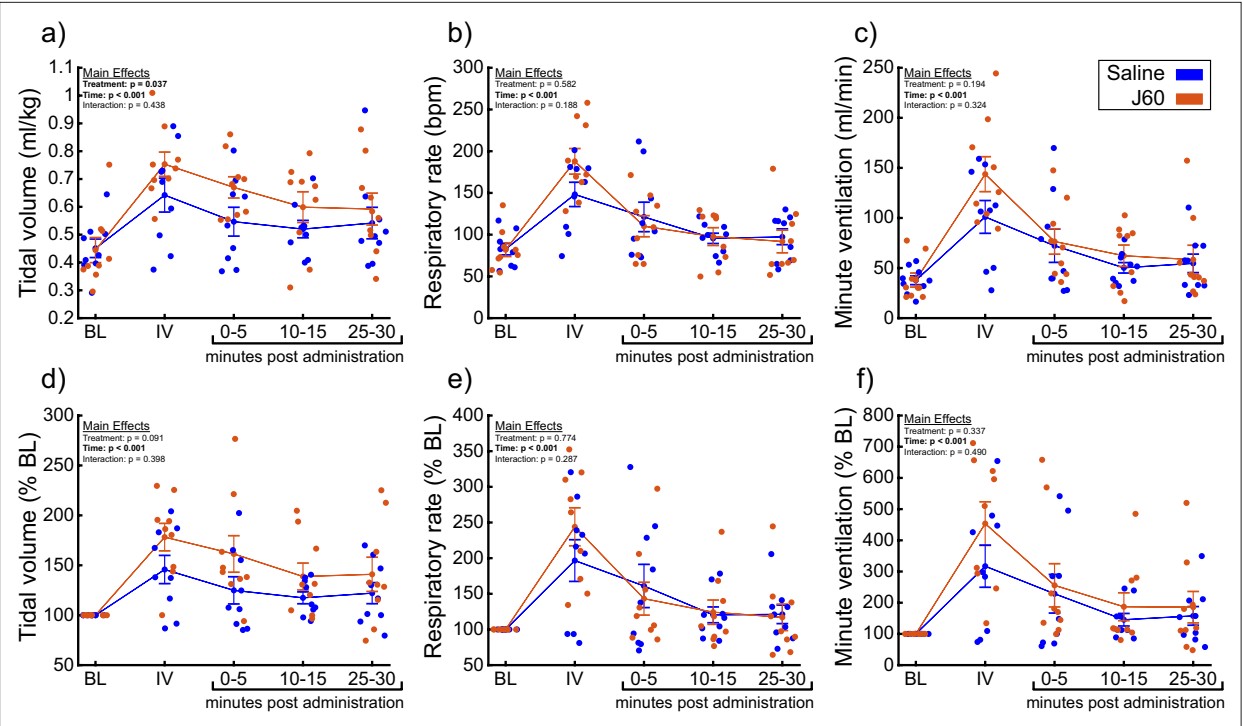

**Figure 5.** DREADD activation increases ventilation in unanesthetized ChAT-Cre rats. Summary plots (*n* = 9; *n* = 3 females) showing the impact of the J60 DREADD ligand on tidal volume, respiratory rate, and minute ventilation are shown in panels a–c. The normalized values (% of baseline) are shown in panels d–f. The DREADD ligand increased tidal volume compared to sham infusion (saline). Error bars depict ±1 SEM. Data were analyzed using two-way RM ANOVAs with alpha = 0.05. Statistical reports for all panels are provided in *Supplementary file 4*. BL = baseline, IV = intravenous infusion period.

The online version of this article includes the following figure supplement(s) for figure 5:

**Figure supplement 1.** Ventilatory responses to hypercapnic–hypoxic respiratory challenge.

effect of Treatment: p = 0.194; *Figure 5c*; Normalized to Baseline: Main effect of Treatment: p = 0.337; *Figure 5f*; *Supplementary file 4*). Responses to a hypercapnic-hypoxia ventilatory challenge were also assessed (*Figure 5—figure supplement 1*). Tidal volume (Normalized to Weight (ml/kg): p = 0.845; *Figure 5—figure supplement 1a*; Normalized to Baseline: p = 0.643; *Figure 5—figure supplement 1d*), respiratory rate (Respiratory Rate: p = 0.262; *Figure 5—figure supplement 1b*; Respiratory Rate Normalized to Baseline: p = 0.734; *Figure 5—figure supplement 1e*), and minute ventilation (Normalized to Body Weight: p = 0.697; *Figure 5—figure supplement 1c*; Normalized to Baseline: p = 0.912; *Figure 5—figure supplement 1f*) did not differ between J60 versus sham condition during hypercapnic–hypoxic ventilatory challenges.

Phrenic nerve recordings were performed to directly assess the effects of DREADD activation on phrenic output. There was no detectable relationship between time post-AAV injection and phrenic response to DREADD activation (Pearson correlation; left peak-to-peak response: p = 0.215; right peak-to-peak response: p = 0.318).

Application of the J60 ligand caused a rapid, sustained, and bilateral increase in phrenic nerve efferent burst amplitude (left phrenic peak-to-peak amplitude (normalized to baseline): p < 0.001; Right phrenic peak-to-peak amplitude (normalized to baseline): p < 0.001; *Figure 6b, c*; *Supplementary file 5*), whereas saline injection had no impact. The increase in phrenic burst amplitude lasted up to 100-min post-J60 administration, at which point the experiment was terminated. Application of the J60 ligand also resulted in an increase in phrenic tonic activity (left phrenic tonic activity (normalized to baseline): p < 0.001; right phrenic tonic activity (normalized to baseline): p < 0.001; *Figure 6d, e*; *Supplementary file 5*).

Heart rate, systolic and diastolic blood pressure, mean arterial blood pressure (MAP), as well as respiratory rate, were also assessed (*Figure 6f–j*). Application of J60 did not affect heart rate (p = 0.587; *Figure 6h*; *Supplementary file 5*) or respiratory rate (p = 0.282; *Figure 6j*; *Supplementary*

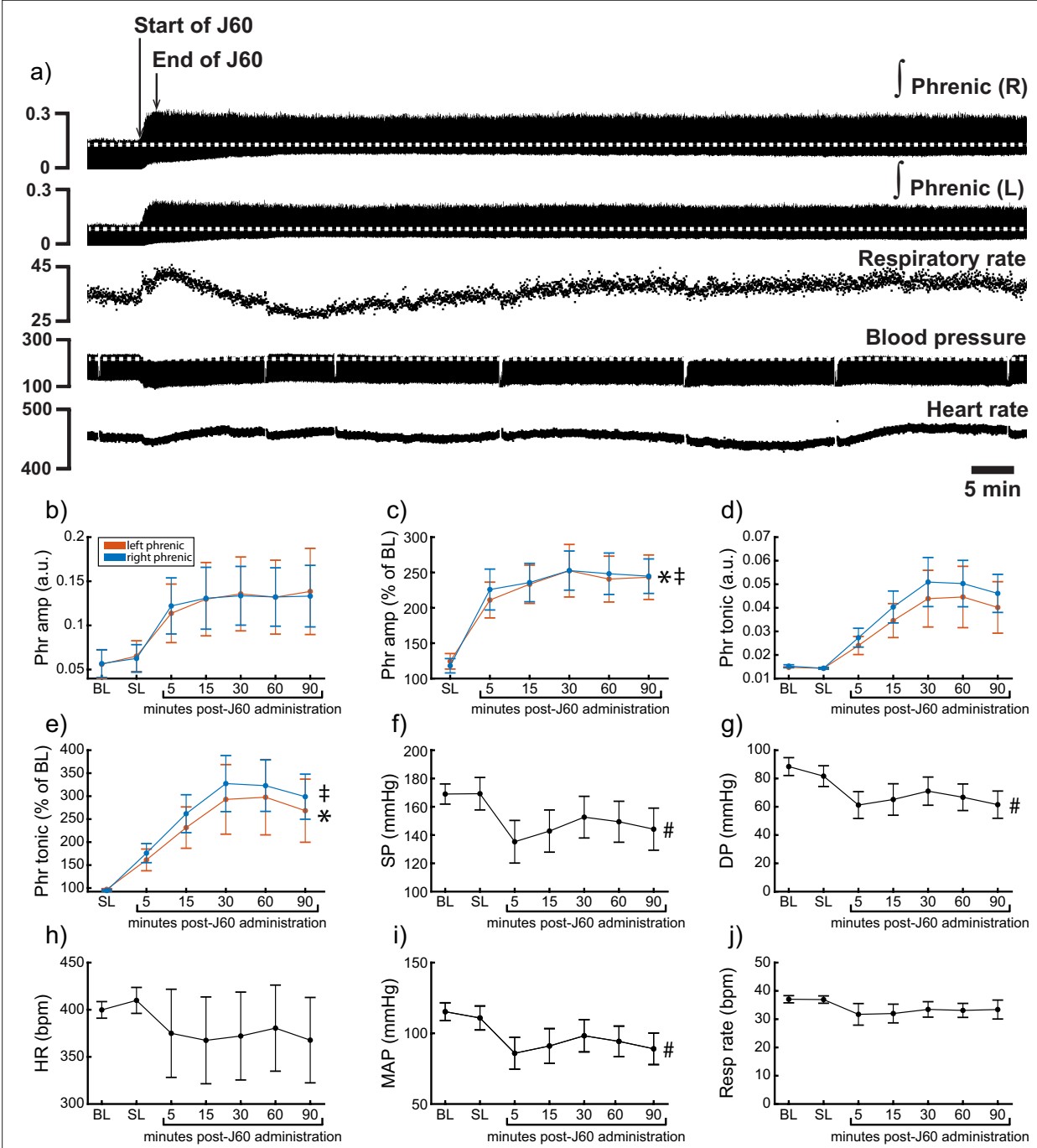

**Figure 6.** DREADD activation increases phrenic nerve output in ChAT-Cre rats. Representative data showing that the J60 DREADD ligand causes a rapid increase in phrenic nerve output (**a**). Mean data (*n* = 9; *n* = 3 females) showing the impact of J60 application on phrenic nerve raw (**b**) and normalized (**c**) peak-to-peak amplitude, raw (**d**) and normalized (**e**) tonic activity, systolic blood pressure (**f**), diastolic blood pressure (**g**), heart rate (**h**), mean arterial blood pressure (**i**), and respiratory rate (**j**). The J60 ligand caused an increase in phrenic peak-to-peak amplitude and tonic activity. Systolic, diastolic, and mean arterial blood pressure all decreased after J60 application. Heart rate and respiratory rate were not statistically different after J60 administration. In panels b–e, the left phrenic is represented in orange, while right phrenic is blue. Error bars depict ±1 SEM. Statistical reports for all panels are provided in *Supplementary file 5*. * and ‡ symbols indicate significant main effects ($p < 0.05$) on one-way RM ANOVA for the left and right hemidiaphragm, respectively. # indicates a significant ($p < 0.05$) effect on one-way RM ANOVA for respiratory rate data. Phr = phrenic, amp = amplitude, BL = baseline, SP = systolic pressure, DP = diastolic pressure, HR = heart rate, MAP = mean arterial pressure.

The online version of this article includes the following figure supplement(s) for figure 6:

**Figure supplement 1.** Impact of DREADD activation on diaphragm electromyogram (EMG) in ChAT-Cre rats.

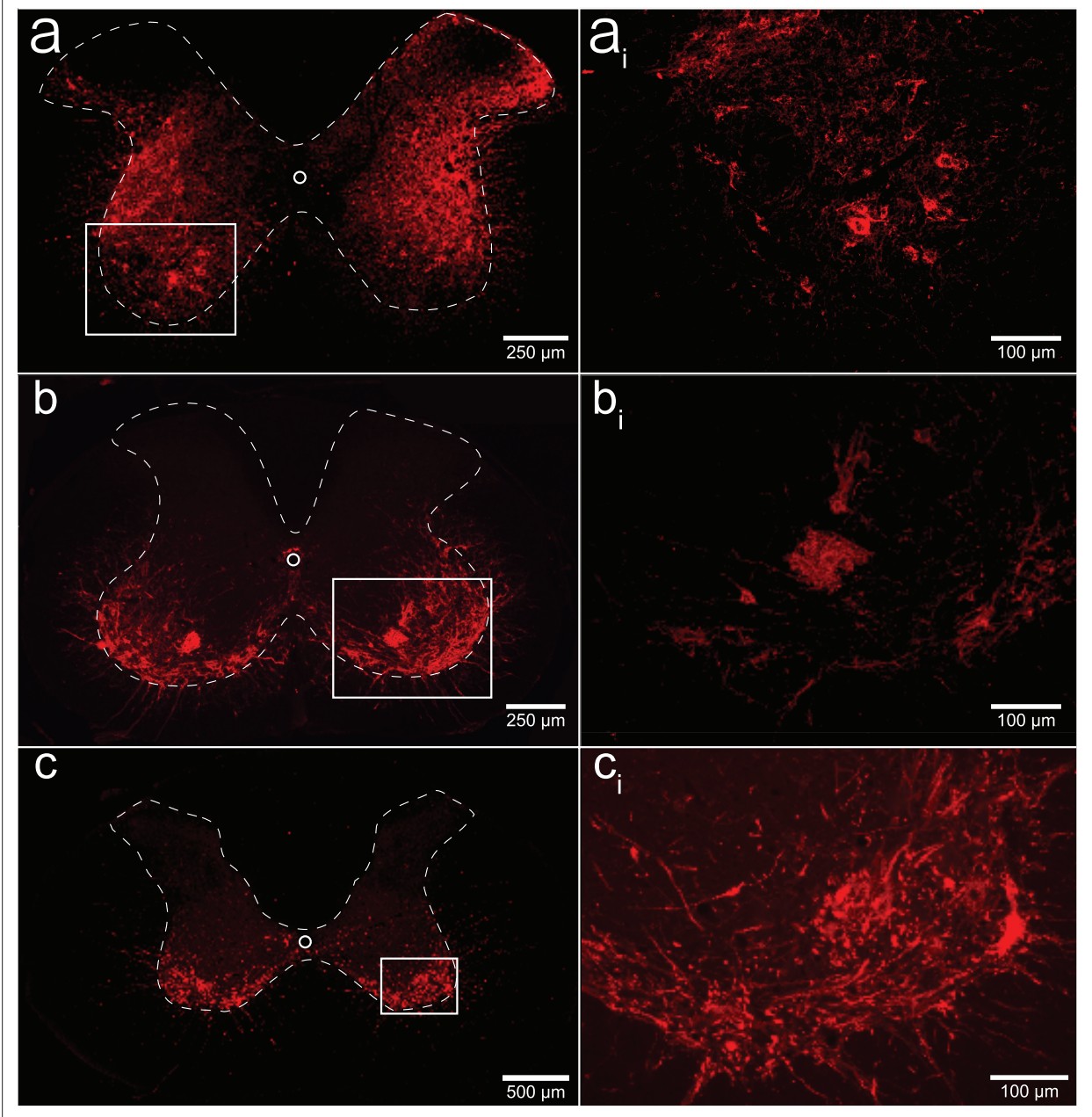

**Figure 7.** Histological assessment of mCherry expression in the C4/C5 spinal segments. Representative photomicrographs of mid-cervical spinal sections from a wild-type mouse (**a, a**$_i$), a ChAT-Cre mouse (**b, b**$_i$), and a ChAT-Cre rat (**c, c**$_i$). Wild-type mice (**a, a**$_i$) showed a non-specific pattern of expression throughout the mid-cervical gray matter. ChAT-Cre mice and rats (**b–c**$_i$) showed expression limited to neurons in the ventral horns. Red color indicates positive and mCherry fluorescence. Dashed white line indicates the approximate white–gray matter demarcation.

file 5) but did result in a decrease in both systolic ($p < 0.001$; *Figure 6f*; *Supplementary file 5*) and diastolic blood pressure ($p < 0.001$; *Figure 6g*; *Supplementary file 5*) as well as MAP ($p < 0.001$; *Figure 6i*; *Supplementary file 5*).

## Histological analysis

We performed a qualitative analysis of the mid-cervical spinal cord from each animal to assess the extent of mCherry fluorophore expression (*Figure 8—figure supplement 1*). All mice from both cohorts showed evidence of mCherry expression in at least one segment of the mid-cervical spinal cord (*Figure 7*) with the exception $n = 1$ ChAT-Cre mouse. This mouse was excluded from all analyses

|  | Wildtype mice (n= 11) | | | | ChAT-Cre mice (n= 9) | | | | ChAT-Cre rats (n= 9) | | | |
|---|---|---|---|---|---|---|---|---|---|---|---|---|
|  | Dorsal | | Ventral | | Dorsal | | Ventral | | Dorsal | | Ventral | |
|  | Left | Right | Left | Right | Left | Right | Left | Right | Left | Right | Left | Right |
| C3 | 4 | 5 | 6 | 6 | 0 | 1 | 3 | 4 | 0 | 1 | 3 | 4 |
| C4 | 7 | 9 | 10 | 9 | 1 | 2 | 5 | 9 | 0 | 1 | 6 | 8 |
| C5 | 11 | 11 | 11 | 11 | 0 | 1 | 6 | 9 | 1 | 1 | 5 | 9 |
| C6 | 10 | 10 | 9 | 9 | 2 | 2 | 5 | 8 | 0 | 0 | 1 | 2 |

**Figure 8.** Qualitative assessment of mCherry expression in the mid-cervical spinal cord. Spinal segments C3–C6 were assessed in quadrants broken into dorsal, ventral, left, and right. Spinal segments were counted as 'positive' if they showed any evidence of mCherry expression in neuronal soma or fibers. The counts therefore indicate the number of animals of a given cohort that were mCherry positive for a given spinal segment quadrant. All animals showed a slight trend for more mCherry expression moving rostral to caudal and for more expression in the ventral versus the dorsal lamina. This trend was more prominent in the ChAT-Cre animals. At the bottom of the table, a heatmap is provided for easier assessment of the distribution of positive mCherry counts across quadrants and spinal segments.

The online version of this article includes the following figure supplement(s) for figure 8:

**Figure supplement 1.** Example quantification of mCherry expression in the C3–C6 spinal cord.

based on a priori exclusion criteria, which stipulated animals must show evidence of mCherry expression in the gray matter of at least one spinal segment from C3 to C6 to be included in the final analysis. A summary of the results is given in *Figure 8*.

Patterns of expression were relatively homogenous in wild-type animals. In this cohort, the number of mice with positive mCherry expression in the gray matter increased on the rostral–caudal axis. Positive mCherry counts were comparable on both the dorsal–ventral and left–right axes, with a majority of mice expressing mCherry in all four quadrants. The diaphragm EMG responses to J60, on average, exhibited similarity between the left and right hemidiaphragm in these mice, aligning with the observed pattern of mCherry expression. (*Figure 1d–f*).

In contrast, mCherry expression in the ChAT-Cre mice cohort was more prevalent in the ventral horns and the right side of the cord. Like the wild-type mice, there was a slight trend for increased mCherry expression moving rostral to caudal. Clear mCherry expression was detectable in the spinal cord in of all nine ChAT-Cre mice included in the final dataset. One additional ChAT-Cre mouse was excluded from analysis as it showed no evidence of mCherry in the mid-cervical spine. Interestingly, this particular mouse appeared to show a modest increase in diaphragm output in response to the J60 ligand in the left hemidiaphragm only (~45% increase compared to baseline activity). While this animal was ultimately excluded from our analysis, it is possible that this mouse did express hM3D(Gq) in the mid-cervical spinal cord but an issue in tissue processing resulted in an inability to visualize the mCherry fluorophore in the spinal tissue. All other ChAT-Cre mice showed robust mCherry expression in the ventral horns of at least one spinal segment from C3 to C6. These mice demonstrated a larger average DREADD response in the right hemidiaphragm than the left (*Figure 2d–f*), possibly stemming from the fact that a greater number of mice exhibited mCherry expression on the right side compared to the left (*Figure 2—figure supplement 1*).

ChAT-Cre rats showed expression predominately in the ventral horns throughout the mid-cervical spinal cords with the highest levels of expression in spinal segments C4 and C5. Expression in this cohort was slightly more prominent on the right side of the cord and in the ventral horns. These histological findings were consistent with the physiological results. Although the magnitude of DREADD response between the left and right phrenic nerves for this cohort was not statistically different, there was a trend of slightly higher right phrenic tonic response compared to the left (*Figure 6b–e*). This trend is mirrored in the pattern of mCherry expression, where expression levels were approximately equal between spinal segments but tended to be slightly higher in the right ventral horns compared to the left.

## Discussion

We describe a novel method to increase diaphragm EMG output by expressing the excitatory DREADD, hM3D(Gq), in the mid-cervical spinal cord, targeting phrenic motoneurons. Following AAV-driven expression of the DREADD in the spinal cord, application of the J60 ligand caused sustained increases in diaphragm output as measured through EMG in spontaneously breathing animals. This response was also verified using direct recordings of phrenic nerve discharge. Additionally, the DREADD ligand was able to produce an increase in inspiratory tidal volume in awake, freely behaving animals. These proof-of-concept studies provide a foundation for further development of this technology toward clinical application for restoring diaphragm activation in conditions such as cervical spinal cord injury.

### Targeted gene delivery to the phrenic motor pool

The intraspinal AAV delivery used here was based on previous studies demonstrating successful gene delivery to phrenic motoneurons (*Alilain et al., 2008*; *Li et al., 2014*; *Li et al., 2015*; *Qiu et al., 2012*). For example, mid-cervical spinal injections of an AAV5 vector encoding the lysosomal enzyme acid alpha-glucosidase (GAA) in animals with Pompe disease (*GAA* null) effectively restores spinal GAA enzyme activity (*Qiu et al., 2012*). Spinal-delivered viral vectors have also been used to successfully drive local expression of channelrhodopsin-2 to enable light activation of diaphragm output (*Alilain et al., 2008*), and to drive expression of the astrocyte glutamate transporter GLT1 in the area of the phrenic motor nuclei (*Li et al., 2014*). Other methods that have been employed to drive gene expression in phrenic motoneurons include intrapleural- and intramuscular diaphragm injection of viral vectors (*Thakre et al., 2023*). Intrapleural delivery requires microinjection to the 'pleural space' between the visceral pleura that lines the lungs and the parietal pleura that covers the thoracic cavity. This technique (*Mantilla et al., 2009*) effectively targets phrenic motoneurons in rodent models of cervical spinal cord injury (*Gransee et al., 2013*; *Martínez-Gálvez et al., 2016*; *Gransee et al., 2017*) and Pompe disease (*Keeler et al., 2020*). Intramuscular diaphragm injection allows the vector to enter phrenic nerve terminals and reach phrenic motoneuron soma via retrograde movement (*Thakre et al., 2023*). Direct diaphragm injection allows for a relatively high target specificity, with the gene of interest expressed almost exclusively in phrenic motoneurons (although expression can also occur in diaphragm myofibers, depending on the promoter sequence used). In pilot experiments, we tested intrapleural and intramuscular diaphragm injections using AAV9 vectors encoding GFP (AAV9-CAG-GFP) or DREADD (AAV9-hSyn-HA-hM3D(Gq)-mCherry and AAV9-hSyn-DIO-hM3D(Gq)-mCherry). We did not, however, observe histological or physiological evidence of phrenic motoneuron transduction with these AAV9 vectors. Direct intraspinal injection (*Li et al., 2014*; *Qiu et al., 2012*) was therefore used to introduce the hM3D(Gq) into the phrenic motor nucleus. While this enabled proof-of-concept for targeting DREADDs to the cervical spinal cord and phrenic motoneurons, the intrapleural or diaphragmatic injection delivery routes might ultimately prove better for selective phrenic motoneuron targeting. We predict that using different AAV serotypes or viruses with better retrograde movement (e.g., 'AAV retro') could optimize the targeting of phrenic motoneurons (*Tervo et al., 2016*).

### DREADD-mediated motoneuron activation

DREADD technology is widely used for studying brain and spinal cord neuronal networks (*Smith et al., 2021*; *Roth, 2016*). Relatively few studies, however, have examined if and how DREADDs can be used to activate (or inhibit) lower motoneurons. Regarding the spinal cord, we are aware of only a few prior publications (*Ouali Alami et al., 2020*; *Saxena et al., 2013*; *Jaiswal et al., 2018*; *Jaiswal and English, 2017*). Two of these studies used pharmacologically selective actuator module, a type of ionotropic chemogenetic receptor, to activate lumbar (*Ouali Alami et al., 2020*; *Saxena et al., 2013*) motoneurons, in mouse models of amyotrophic lateral sclerosis. In the remaining studies, excitatory DREADDs were applied to spinal motoneurons to improve axon regeneration following peripheral nerve injury (*Jaiswal et al., 2018*; *Jaiswal and English, 2017*). A small but growing body of work has employed DREADDs to activate hypoglossal (XII) motoneurons in the brainstem (*Doyle et al., 2021*). Collectively, these studies show that once hM3D(Gq) is expressed in XII motoneurons, DREADD ligands will rapidly produce an increase in the EMG activation of tongue muscles (*Fleury Curado et al., 2021*; *Singer et al., 2022*). This increase in tongue muscle output tends to mani-fest as an increase in the inspiratory-related activation and tonic discharge across the respiratory

cycle. Since increased tongue muscle activation can promote patency of the upper airway, XII motoneuron DREADD expression has been suggested as a possible treatment for obstructive sleep apnea (*Fleury Curado et al., 2017*; *Fleury Curado et al., 2021*; *Horton et al., 2017*). For the present study, the primary innovation is the first application of DREADD technology to phrenic motoneurons. This approach was highly effective at driving sustained activation of the diaphragm muscle. The underlying mechanisms are discussed next.

## Chemogenetic stimulation of breathing

An important consideration is how DREADD-induced increases in the excitability of spinal neurons, including phrenic motoneurons, interacts with the endogenous neural control of breathing. Phrenic motoneurons receive a rhythmic, monosynaptic, glutamatergic synaptic input from medullary neurons. Acting via NMDA and AMPA receptors, this produces phrenic motoneuron depolarization and subsequent diaphragm muscle contraction (*Fuller et al., 2022*). Activating DREADDs on phrenic motoneurons should lower the threshold for activation via excitatory glutamatergic synaptic inputs, which would produce a greater output during the inspiratory phase. Alternatively, DREADD activation could directly lead to phrenic motoneuron action potentials even in the absence of synaptic input from the brainstem. This latter possibility could explain the tonic discharge (i.e., EMG output across the entire respiratory cycle) that was noted to occur after delivery of the DREADD ligand. Non-specific spinal cord DREADD expression, as occurred in the wild-type mice (e.g., *Figure 7a, ai*), would likely produce an increase in the excitability and/or activation of phrenic motoneurons as well as propriospinal neurons in the immediate vicinity. Neurophysiological (*Streeter et al., 2020*), as well as anatomical data (*Lane, 2011*; *Lane et al., 2008*), confirm synaptic connections between mid-cervical interneurons and phrenic motoneurons, making it possible that DREADD activation of these interneurons impacted the diaphragm motor response in the wild-type mice.

The control of breathing is also impacted by well-established 'closed loop' physiologic feedback mechanisms regulating lung volume and arterial blood gases (*Molkov et al., 2017*; *Dempsey and Welch, 2023*). For example, if DREADD-induced activation of the diaphragm leads to increased alveolar ventilation, and metabolic rate is not impacted, then arterial $CO_2$ values will decrease and the overall neural drive to breathe will also decrease. Vagal afferent feedback corresponding to increased lung volume also has a powerful inhibitory impact on inspiration and therefore diaphragm activation. However, the sustained increase in diaphragm EMG and tidal volume that we observed following application of the DREADD ligand indicates that these mechanisms, if activated, were not sufficient to fully inhibit the increased phrenic motoneuron output. In this regard, our additional experiments in which direct recordings of bilateral phrenic nerve discharge are informative. These nerve recording experiments were done to enable direct evaluation of the impact of spinal DREADD activation on phrenic motor output while keeping arterial blood gases and lung volume constant. Under these more rigorously controlled conditions, IV delivery of the DREADD ligand produced a rapid and sustained increase in inspiratory burst amplitude in the phrenic nerve, and with no impact on the rate of the inspiratory bursts. The relative increase in inspiratory motor output was considerably greater in the phrenic nerve recording experiments (~250% of baseline) as compared to the diaphragm EMG response in spontaneously breathing animals (~100% of baseline). This may indicate that vagal and/or blood gas-related inhibitory mechanisms, as mentioned above, somewhat constrained the response to the DREADD ligand in the spontaneously breathing animal.

## Critique of methods

There are a few caveats that should be discussed. First, the precision of the AAV delivery could be improved by further refining spinal injection surgical techniques. In the current study, we used a stereotaxic frame and previously validated coordinates (*Qiu et al., 2012*; *McGuire et al., 2005*) to guide the intraspinal AAV injections. However, we observed variability in the laterality (i.e., left vs. right side of the spinal cord) of mid-cervical mCherry expression as well as the physiological response to the DREADD ligand, particularly in the ChAT-Cre mice (e.g., *Figure 2* and *Figure 8*). This could have occurred due to subtle variations of the positioning of the animal within the stereotaxic frame, and/or placement of the needle tip, leading to slight deviations for the desired coordinates between the left and right phrenic nuclei. Second, we did not unequivocally verify that the DREADD was expressed in phrenic motoneurons using retrograde labeling methods (*Mantilla et al., 2009*; *Rana et al., 2022*).

However, the phrenic motor nucleus has been well described in the mouse (*Qiu et al., 2010*) and the rat (*Rana et al., 2020*; *Rana et al., 2019*), and the fluorophore (mCherry) expression observed in our experiments is very clearly in the expected location of phrenic motoneurons (*Figure 7b–ci*; *Figure 8— figure supplement 1*, *Figure 2—figure supplement 1*). Further, the robust increase in phrenic motor output after the DREADD ligand, particularly in the ChAT-Cre rat experiment (*Figure 6*) is further evidence of effective phrenic motoneuron targeting.

## Conclusion

Our data support the conclusion that cervical spinal cord directed chemogenetic receptors can be used to produce sustained increases in phrenic motor output, diaphragm activation, and inspiratory tidal volume. Collectively, the data indicate that DREADDs should be directed exclusively to phrenic motoneurons versus non-specific expression in the immediate vicinity. In this regard, improvement of the AAV delivery methods will increase the selectivity of the approach for more precise targeting of phrenic motoneurons. Concerning the 'translational value' of this work, spinal cord chemogenetics may have application to clinical conditions associated with an inability to activate the diaphragm. For example, incomplete cervical spinal cord injury is a condition in which the bulbospinal synaptic inputs to phrenic motoneurons are interrupted. After incomplete cervical spinal cord injury, focal expression of an excitatory DREADD in phrenic motoneurons could be used to increase the excitability of these cells, thereby improving the efficacy of spared bulbospinal synaptic inputs which convey 'inspiratory drive'.

## Methods

### Key resources table

| Reagent type (species) or resource | Designation | Source or reference | Identifiers | Additional information |
|---|---|---|---|---|
| Recombinant DNA reagent | pAAV-hSyn-hM3D(Gq)-mCherry | Addgene | RRID:Addgene_50474 | AAV transgene plasmid |
| Recombinant DNA reagent | pAAV-hSyn-DIO-hM3D(Gq)-mCherry | Addgene | RRID:Addgene_44361 | AAV transgene plasmid |
| Chemical compound, drug | JHU37160 | HelloBio | HB6261 | DREADD agonist |
| Software, algorithm | MATLAB | MathWorks | Version R2019a (9.6.1072779) RRID:SCR_001622 | |
| Software, algorithm | SigmaPlot | Systat Software | Version 14 RRID:SCR_003210 | |
| Software, algorithm | R | The R Foundation for Statistical Computing | Version 4.3.1 – 'Beagle Scouts' RRID:SCR_001905 | |

### Animals

Experiments were carried out using C57Bl/6, wild-type mice (Taconic), ChAT-Cre mice (B6.129S6-Chattm2(cre)Lowl/J; Jackson Laboratories), and ChAT-Cre rats (LE-Tg(Chat-Cre)5.1Deis; Rat Resource & Research Center). Animals were singly housed in a controlled environment (12 hr light–dark cycle) with food and water ad libitum. All experiments were conducted in accordance with the NIH Guidelines Concerning the Care and Use of Laboratory Animals and were approved by the University of Florida Institutional Animal Care and Usage Committee (protocol #202107438). A full experimental timeline for mouse and rat experiments is shown in *Figure 9*, panels a and b, respectively.

### Adeno-associated viral vectors

All animals underwent intraspinal injections (see section below) of an adeno-associated viral vector (AAV) encoding the excitatory DREADD (hM3D(Gq)) under a human synapsin promoter. Wild-type mice received injections of AAV9-hSyn-HA-hM3D(Gq)-mCherry (titer: $2.44 \times 10^{13}$ vg/ml) while ChAT-Cre mice and rats received injections of a similar construct with a double-floxed inverted open-reading frame (DIO) allowing for Cre-dependent transgene expression (AAV9-hSyn-DIO-hM3D(Gq)-mCherry; titer: $2.07 \times 10^{12}$ vg/ml). The pAAV-hSyn-hM3D(Gq)-mCherry (Addgene plasmid # 50474; http://n2t.

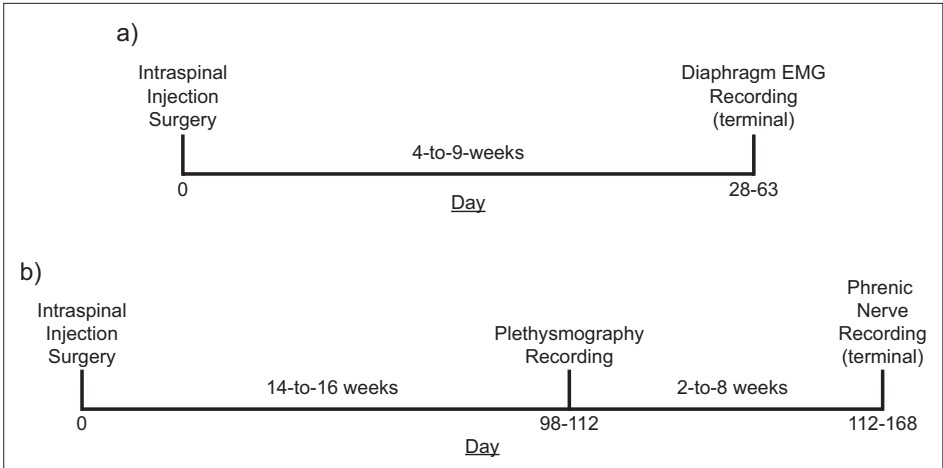

**Figure 9.** Experimental timelines. Timelines of mouse (**a**) and rat (**b**) studies. Both cohorts of animals underwent an initial surgery to introduce an AAV vector encoding the excitatory DREADD, hM3D(Gq) in the ventral mid-cervical spinal cord bilaterally. Mice incubated for 4–9 weeks before undergoing terminal diaphragm electromyogram (EMG) recordings. Rats incubated for 14–16 weeks before undergoing plethysmography recordings, 2–8 weeks later rats underwent terminal phrenic nerve recordings. In all experiments, baseline parameters were established followed by an infusion of vehicle, and subsequently the selective DREADD ligand, J60.

net/addgene:50474; RRID:Addgene_50474) and pAAV-hSyn-DIO-hM3D(Gq)-mCherry (Addgene plasmid # 44361; https://n2t.net/addgene:44361; RRID:Addgene_44361) transgene plasmids were gifts from the laboratory of Dr. Brian Roth at the University of North Carolina. Viral preparations were generated and titered by the University of Florida Powell Gene Therapy Center Vector Core Lab. Vectors were purified by iodixanol gradient centrifugation and anion-exchange chromatography as previously described (*Zolotukhin et al., 2002*).

## Intraspinal injections

An AAV encoding the gene for the excitatory DREADD, hM3D(Gq) was delivered to the mid-cervical spinal cord. Mice were 6–10 weeks old (wild-type cohort: 7–9 weeks; ChAT-Cre cohort: 6–10 weeks) at the time of injection while ChAT-Cre rats were 2–5 months old. Surgery was performed under aseptic conditions. Mice were anesthetized with isoflurane (induction: 3–4% isoflurane; maintenance: 2–3% isoflurane in 100% $O_2$) while rats were anesthetized with a mixture of ketamine (100 mg/kg) and xylazine (10 mg/kg) delivered intraperitoneally. Animals were placed prone on a circulating water heating pad to maintain body temperature. A longitudinal incision was made starting at the base of the skull and extending caudally. The underlying back musculature was opened from the base of the skull to spinal segment C6. Using a micro-curette, the muscle and connective tissue overlying laminae C3–C5 were removed. A laminectomy of the C4 dorsal lamina exposed the dura mater below. A bilateral durotomy was then performed exposing the spinal cord. A Hamilton syringe (34-gauge needle) held in a Kopf stereotaxic frame was used to inject 1 µl of AAV9-hSyn-DIO-hM3D(Gq)-mCherry (ChAT-Cre mice and rats) or AAV9-hSyn-HA-hM3D(Gq)-mCherry (C57Bl/6 mice), bilaterally into the ventral horns at C4. Injections were made 0.5 mm lateral to the spinal midline at a depth of 0.9 mm for mice (*Qiu et al., 2012*) and 1 mm lateral to midline at a depth of 1.5 mm for rats (*McGuire et al., 2005*.) The needle was left to dwell for 5 min. Following injections, the overlying muscle and fascia were sutured with absorbable suture, the skin closed, and the animal returned to its home cage. Animals received a post-operative analgesia regiment of subcutaneous buprenorphine (1 mg/kg; slow-release formulation) and carprofen (5 mg/kg) for the first 3 days after surgery.

## Diaphragm EMG recordings

Recordings were conducted using wild-type (n = 11; n = 7 females) and ChAT-Cre mice (n = 9; n = 6 females; n = 1 excluded from analysis), 4–9 weeks following intraspinal injections of AAV-DREADD. Mice were anesthetized with 2–3% isoflurane in a closed chamber and then placed supine on a closed

loop heating pad to maintain rectal temperature at 37 ± 0.5°C (model 700 TC-1000, CWE Inc). Mice spontaneously inhaled 2% isoflurane in 100% $O_2$ for the duration of the experiment.

A laparotomy was performed and two sets of 50 μm tungsten wires were placed in the mid-costal region of the left and right hemidiaphragm. The tips of each wire were de-insulated, bent into small hooks, and inserted through the diaphragm approximately 3 mm apart. The recorded EMG signals were amplified (1000×) and filtered (100–1000 Hz) using a differential amplifier (A-M systems model 1700). Signals were digitized at 10 kS/s using a Power 1401 (CED, Cambridge, UK).

Once a stable plane of anesthesia was reached, mice underwent a 10-min recording to establish baseline diaphragm EMG parameters. Subsequently, mice received injections of vehicle (100 μl of saline delivered intraperitoneally) followed by a 20-min recording. Mice then received an intraperitoneal injection of the selective DREADD agonist, JHU37160 (J60; 0.1 mg/kg, HB6261, HelloBio), and recordings continued for 90 min. At the conclusion of each experiment, mice underwent transcardial perfusion with saline followed by 4% paraformaldehyde. Following perfusion, spinal cords were harvested for histological analysis.

## J60 control experiments

A small cohort of animals ($n$ = 2 C57/bl mice; $n$ = 3 Sprague Dawley rats) was used to assess the impact of J60 (0.1 mg/kg) on diaphragm EMG activity in the absence of hM3D(Gq) expression. The animals used in this experimental include $n$ = 2 C57/bl mice that had undergone intrapleural injection (i.e., injection to the thoracic cavity) of an AAV9 construct encoding the red fluorescent protein, mCherry and $n$ = 3 vector naive Sprague Dawley rats.

Recordings in mice proceeded as described above (see *Diaphragm EMG recordings*). In rat recordings, rats were induced with 3% isoflurane in 100% $O_2$ and moved onto a closed-loop heating pad set to maintain rectal temperature at 37 ± 1°C (model 700 TC-1000, CWE Inc). Rats were tracheotomized and ventilated (Model 683; Harvard Apparatus Inc) with a gas mixture of 50% $O_2$, and 1% $CO_2$, balanced with $N_2$. End-tidal $CO_2$ was maintained at 45–47 mmHg throughout the experimental protocol (Capnogard; Novametrix). Rats were converted from isoflurane to urethane anesthesia (2.1 g/kg at 6 ml/hr; IV). At the competition of urethane dosing, lactated Ringer's was administered (2 ml/hr; IV) to keep the animal hydrated and ensure the catheter remained viable for J60 administration. A femoral artery catheter (polyethylene tubing; PE 50; Intramedic) was placed to enable monitoring of arterial blood pressure via a transducer amplifier (TA-100, CWE).

At the beginning of the experimental period, rats underwent a 10-min recording to establish baseline diaphragm EMG parameters. This was followed by an IV injection of vehicle (0.6 ml of saline) and a subsequent 20-min recording. Next, rats received an IV infusion of the J60 agonist (0.1 mg/kg), and the recording continued for 90 min. At the end of the experiment, rats were euthanized via an overdose of pentobarbital sodium and phenytoin sodium (150 mg/kg) given intravenously. Death was confirmed by thoracotomy once breathing had ceased, and a heartbeat was no longer detectable.

## Whole-body plethysmography

ChAT-Cre rats ($n$ = 9; $n$ = 3 females) were studied using flow-through whole-body plethysmography 14–16 weeks after intraspinal delivery of AAV9-hSyn-DIO-hM3D(Gq)-mCherry, as described above. A tail vein catheter was placed to allow for IV infusion of the J60 ligand and vehicle. An IV catheter was externalized via a port in the plethysmograph allowing for IV infusion during recording without handling the animal or opening the plethysmograph. Unanesthetized rats were sealed into the Plexiglas plethysmograph with airflow maintained at 6 l/min for the duration of the recording. The recording protocol consisted of a 40-min acclimation period (inspired air: 21% $O_2$, 79% $N_2$), followed by a 7-min ventilatory challenge (10% $O_2$, 7% $CO_2$, 83% $N_2$) and a 10-min normoxic recovery period (21% $O_2$, 79% $N_2$). Subsequently, rats underwent a 20-min long, pre-vehicle, baseline under normoxic conditions (21% $O_2$, 79% $N_2$) followed by a 2-min-long IV infusion of the J60 vehicle (saline; 0.6 ml). Following vehicle infusion recording continued for 30 min followed by a 7-min ventilatory challenge (10% $O_2$, 7% $CO_2$, 83% $N_2$) and 10 min of normoxic breathing (21% $O_2$, 79% $N_2$). After an additional 20-min pre-J60 baseline (21% $O_2$, 79% $N_2$), an IV infusion of the J60 ligand was given (0.1 mg/ml dose; 2 min long; final volume standardized to 0.6 ml) and recordings continued for 30 min followed by a final ventilatory challenge (10% $O_2$, 7% $CO_2$, 83% $N_2$). The ventilatory challenges were performed to assess the ability to increase breathing.

## Phrenic nerve recordings

Two to eight weeks following plethysmography recordings, bilateral phrenic nerve recordings were performed. This procedure was done to directly assess the effect of DREADD activation on phrenic motor output under rigorously controlled experimental conditions. Anesthesia was induced by placing the rat in a closed chamber to inhale 3% isoflurane in 100% $O_2$. Rats were then moved onto a closed-loop heating pad set to maintain rectal temperature at 37 ± 1°C (model 700 TC-1000, CWE Inc). Isoflurane anesthesia was maintained using a nose cone. Once a surgical plane of anesthesia was reached as evidenced by loss of corneal reflexes and hindlimb withdrawal, rats were tracheotomized and ventilated (VentElite, model 55-7040; Harvard Apparatus Inc) with a gas mixture of 50% $O_2$, 1% $CO_2$, balanced with $N_2$. End-tidal $CO_2$ was maintained at 45–47 mmHg throughout the surgery and experimental protocol (Capnogard; Novametrix). Ventilator frequency was maintained between 65 and 75 breaths/min, and tidal volume was set at 7 ml/kg (*Lee et al., 2015*). The vagus nerves were transected bilaterally to prevent entrainment of phrenic efferent output with the ventilator.

A tail vein catheter was placed to allow for IV infusion of urethane anesthesia, supplementary fluids, and the J60 ligand. Rats were slowly converted from inhaled isoflurane to urethane anesthesia (2.1 g/kg at 6 ml/hr; IV). During this conversion, the depth of anesthesia was consistently monitored by evaluating the pedal withdrawal reflex. Following administration of the full urethane dose, a mixture of 8.4% sodium bicarbonate and lactated Ringer's was administered (2 ml/hr; IV) to maintain acid–base balance. To prevent movements and EMG contamination of the phrenic neurogram pancuronium bromide was administered (3 mg/kg IV, Sigma-Aldrich, St Louis) to achieve neuromuscular blockade. A catheter (polyethylene tubing; PE 50; Intramedic) was placed in the femoral artery to enable monitoring of arterial blood pressure via a transducer amplifier (TA-100, CWE) and allow withdrawal of arterial blood samples (65 µl) for measurement of partial pressure of $CO_2$ (PaCO$_2$) and $O_2$ (PaO$_2$), pH, and base excess (ABL 90 Flex, Radiometer; Copenhagen, Denmark).

The phrenic nerves were exposed bilaterally using a dorsal approach as described previously (*Thakre and Fuller, 2024*; *Thakre et al., 2021*). Briefly, a midline incision was made at the base of the skull extending to spinal level T2. The muscles connecting the shoulder blades to the spinal column were separated to expose the phrenic nerves. The phrenic nerves were isolated, cut distal to the spinal cord, and suctioned into custom-made glass electrodes filled with 0.9% saline solution. Phrenic nerve activity was amplified (×10 kHz) using a differential AC amplifier (Model 1700, A-M systems, Everett, WA), band-pass filtered (100 Hz to 3 kHz), and digitized at 25 ks/s (Power 1401, CED).

At the beginning of the experiment, the apneic threshold was determined by slowly reducing the inspired $CO_2$ until phrenic nerve inspiratory activity ceased for 60 s. The recruitment threshold was established by slowly increasing the inspired $CO_2$ until phrenic bursting returned. The end-tidal $CO_2$ (ETCO$_2$) was then maintained 2–3 mmHg above the recruitment threshold for the duration of the experiment. After achieving a stable phrenic nerve recording and blood gases a 15-min-long baseline recording was collected (50% $O_2$, 3% $CO_2$) followed by a brief, 5-min exposure to hypoxia (11.5% $O_2$, 3% $CO_2$) and 10- to 15-min recovery period (50% $O_2$, 3% $CO_2$). Subsequently, IV infusion of vehicle (saline) was given followed by a 15-min recording period. The J60 ligand (0.1 mg/kg) was then administered intravenously over a 2-min infusion period followed by a 100-min recording period.

Arterial blood samples were collected at specific intervals: initially at baseline, during the last minute of each hypoxia episode, 15-min post-vehicle administration, and subsequently at 20-, 40-, 60-, 80-, and 100-min post-J60 administration. Baseline blood gas values served as references to assess if further arterial samples were isocapnic. To keep end-tidal $CO_2$ and PaCO$_2$ near baseline (within ±2.0 mmHg), minor adjustments to inspired $CO_2$ and ventilation rate were made as needed. PaO$_2$ was kept above 150 mmHg, except during hypoxia; if it dropped below, $O_2$ intake was increased by 5%, and a new blood sample was analyzed within 5 min.

At the end of the experiment, rats were exposed to a second 5-min episode of hypoxia (11.5% $O_2$) followed by a brief 'maximal' chemoreceptor challenge induced by switching off the mechanical ventilator until the animal exhibited a 'gasping-like' phrenic discharge pattern (approximately 20–30 s). If the increase in phrenic nerve amplitude in response to the 'maximal' challenge was lower than the response observed during either hypoxic episode, it was considered a sign of deteriorating nerve–electrode contact, and the preparation was excluded from all formal analyses. Rats were then perfused transcardially with heparinized saline followed by 4% paraformaldehyde and spinal cords were harvested for histological analysis.

## Histology

Spinal cords were harvested and placed in 4% paraformaldehyde for 24 hr. The cords were subsequently moved to a cryo-protecting solution (30% sucrose in 1× PBS) for a minimum of 3 days. Cervical and thoracic spinal cords were blocked in optimal cutting temperature media and cryosectioned at 20 μm. The viral constructs included a red fluorescent protein (mCherry) fused to the hM3D(Gq) DREADD which allowed evaluation of DREADD expression by assessing mCherry expression via fluorescence microscopy.

We performed a qualitative assessment of mCherry expression in the mid-cervical spinal cord. One intact section from the middle of each spinal segment (C3–C6) was chosen as a representative section and underwent assessment. Sections were segmented into the following quadrants: left dorsal, right dorsal, left ventral, and right ventral. The quadrant was scored as 'positive' if mCherry-positive neurons or fibers were observed; otherwise, the sub-segment was marked 'negative' (see *Figure 8—figure supplement 1* for example). The entirety of the gray matter from each section was analyzed for all animals, whether wild-type or ChAT-Cre. Although ChAT-Cre expression was expected to be limited primarily to motoneurons, which are the predominant ChAT-positive neuronal subtype in the spinal cord, there is also evidence of ChAT-positive interneuron populations (*Gotts et al., 2016*; *Alkaslasi et al., 2021*; *Mesnage et al., 2011*) which we also wished to capture in our analysis. Results were compiled into a summary table showing the total positive counts by animal cohort, spinal segment, and quadrant (see Results section; *Figure 8*). Animals that showed no positive mCherry labeling in the C3–C6 cord were excluded from analysis.

## Data analysis and statistics

Custom MATLAB (MathWorks; Natick, MA) scripts were created, and are available upon request. These scripts were used to analyze diaphragm EMG, phrenic nerve, and plethysmography waveforms. EMG signals were digitally filtered using a second-order, bandpass Butterworth filter (100–1000 Hz) and then rectified and integrated by taking the absolute value of the signal followed by applying a moving median filter (50 ms time constant for mice; 75 ms time constant for rats) and moving average filter (50 ms time constant for mice; 175 ms time constant for rats). The script identified each EMG burst and calculated peak amplitude, minimum amplitude (tonic activity), and AUC for each burst which was then averaged across animals and compared across experimental conditions.

Phrenic nerve signals were digitally filtered using a second-order, bandpass Butterworth filter (100–3 kHz) and then rectified and integrated by taking the absolute value of the signal followed by applying a moving median filter (50 ms time constant) and moving average filter (50 ms time constant). The analysis script calculated the peak phrenic burst amplitude and minimum amplitude for each burst which was then averaged across animals and compared across experimental conditions. Systolic (SP), diastolic (DP), and mean arterial blood pressure (MAP; formula: $MAP = DP + 1/3 (SP - DP)$) along with instantaneous heart rate were calculated from the arterial pressure trace.

In plethysmography experiments, airflow pressure, chamber temperature, chamber humidity, barometric pressure, and animal body temperature were used to calculate respiratory frequency, tidal volume, and ventilation via a custom MATLAB script. Tidal volume was calculated using the Drorbaugh and Fenn equation (*Drorbaug and Fenn, 1955*).

Statistical analyses were performed using SigmaPlot 14 (Systat Software) and R (The R Foundation for Statistical Computing; version 4.3.1). In mouse diaphragm EMG studies, one-way repeated measure ANOVA was used to statistically compare diaphragm EMG peak amplitude, AUC, tonic activity, and heart rate across time before and after J60 application. Paired *t*-tests were used to compare left and right hemidiaphragm EMG peak amplitude, AUC, tonic activity, and heart rate between ChAT-Cre and wild-type mice at the 30-min post-J60 administration time point. Differences in mortality between wild-type and ChAT-Cre mice post-J60 application were assessed using Pearson's Chi-squared test with Yates' continuity correction using the chisq.test function in R. In instances where animals did not survive the entire duration of the anesthetized recording, data up until the time point preceding their death was included.

In control EMG experiments, one-way repeated measures ANOVA was used to compare EMG peak responses across baseline, sham injection, and J60 administration. These data were also assessed normalized to baseline, in which case EMG peak responses after sham injection and J60 application were compared using paired *t*-tests.

In plethysmography experiments, two-way repeated measures ANOVA was used to compare raw and normalized tidal volume, respiratory frequency, and minute ventilation across time and treatment (saline vs. J60). Paired *t*-tests were used to compare responses to hypercapnic–hypoxic ventilatory challenges across treatments. One-way repeated measures ANOVA was used to compare phrenic peak amplitude, systolic and diastolic blood pressures, mean arterial blood pressure, and respiratory rate across time for phrenic nerve recordings. The relationship between time post-AAV injection and average phrenic response to J60 was assessed for ChAT-Cre rats using the cor.test function in R to run a Pearson's product moment correlation. Both male and female animals were included in this study to improve the generalizability of the results. However, we were not adequately powered for sex comparisons and therefore did not perform any statistical analysis to assess sex differences.

In cases of significant main effects, the Tukey post hoc test was used to assess differences between individual time points. For instances where data failed to meet general linear model assumptions (i.e., normality, homogeneity of variances), nonparametric equivalents of the previously mentioned statistical tests were used. Data were considered statistically significant when p ≤ 0.05. The mean data are presented along with the standard error of the mean.

## Acknowledgements

Support for this work was provided by the National Institutes of Health: 5R01HD052682-14 (DDF) and a Graduate School Funding Award from the University of Florida (ESB).

## Additional information

### Funding

| Funder | Grant reference number | Author |
|---|---|---|
| National Institutes of Health | 5R01HD052682 | David D Fuller |
| Graduate School Funding Award from the University of Florida | | Ethan S Benevides |

The funders had no role in study design, data collection, and interpretation, or the decision to submit the work for publication.

### Author contributions

Ethan S Benevides, Conceptualization, Data curation, Formal analysis, Writing – original draft, Writing – review and editing; Prajwal P Thakre, Sabhya Rana, Conceptualization, Data curation, Formal analysis, Writing – review and editing; Michael D Sunshine, Data curation, Formal analysis, Writing – review and editing; Victoria N Jensen, Data curation, Writing – review and editing; Karim Oweiss, Resources, Methodology, Writing – review and editing; David D Fuller, Conceptualization, Formal analysis, Supervision, Funding acquisition, Validation, Investigation, Project administration, Writing – review and editing

### Author ORCIDs

Ethan S Benevides https://orcid.org/0000-0002-2472-0606
Sabhya Rana https://orcid.org/0000-0002-1303-6614
David D Fuller https://orcid.org/0000-0001-5013-5972

### Ethics

All procedures described in this manuscript involving rats, mice, and tissue were approved by the University of Florida Institutional Animal Care and Use Committee (protocol #202107438) and in strict accordance with the US National Institute of Health (NIH) Guide for the Care and Use of Laboratory Animals.

Reviewer #1 (Public review): https://doi.org/10.7554/eLife.97846.3.sa1

Reviewer #2 (Public review): https://doi.org/10.7554/eLife.97846.3.sa2
Author response https://doi.org/10.7554/eLife.97846.3.sa3

## Additional files

### Supplementary files
Supplementary file 1. Statistical summary for the impact of DREADD activation on diaphragm EMG in wild-type mice. Time points are in reference to minutes passed since J60 infusion. Summary data are presented in *Figure 1*. EMG = electromyography, AUC = area under the curve, RM = repeated measures, df = degrees of freedom. Bolded p-values indicate p < 0.05.

Supplementary file 2. Statistical summary for the impact of DREADD activation on diaphragm EMG in ChAT-Cre mice. Time points are in reference to minutes passed since J60 infusion. Summary data are presented in *Figure 2*. EMG = electromyography, AUC = area under the curve, RM = repeated measures, df = degrees of freedom. Bolded p-values indicate p < 0.05.

Supplementary file 3. Statistical summary for the impact of DREADD activation on diaphragm EMG in wild-type mice versus ChAT-Cre mice at the 30-min post-J60 infusion time point. Summary data are presented in *Figure 3*. EMG = electromyography, AUC = area under the curve, df = degrees of freedom. Bolded p-values indicate p < 0.05.

Supplementary file 4. Statistical summary for the impact of DREADD activation on plethysmography outcomes in unanesthetized ChAT-Cre rats using two-way repeated measures ANOVAs. Each outcome measure is presented normalized to body weight (with the exception of respiratory rate) and normalized to values at baseline. Summary data are presented in *Figure 5*. RM = repeated measures, ml/kg = milliliters of air per kilogram of animal's body weight, df = degrees of freedom. Bolded p-values indicate p < 0.05.

Supplementary file 5. Statistical summary for the impact of DREADD activation on phrenic nerve activity in ChAT-Cre rats. Time points are in reference to minutes passed since J60 infusion. Summary data are presented in *Figure 6*. RM = repeated measures, df = degrees of freedom. Bolded p-values indicate p < 0.05.

MDAR checklist

### Data availability
All data generated as a part of this study is available at https://doi.org/10.17605/OSF.IO/ZB9CG. MATLAB code used to analyze raw data and produce figures is publicly available at https://github.com/ESBenevid/chemogenetic-phrenic-stimulation (copy archived at *Benevides, 2025*).

The following dataset was generated:

| Author(s) | Year | Dataset title | Dataset URL | Database and Identifier |
|---|---|---|---|---|
| Benevides ES, Thakre PP, Rana S, Sunshine MD, Jensen VN, Oweiss K, Fuller DD | 2025 | Benevides et al. 2025: Chemogenetic stimulation of phrenic motor output and diaphragm activity | https://doi.org/10.17605/OSF.IO/ZB9CG | Open Science Framework, 10.17605/OSF.IO/ZB9CG |

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
