## [Editor Report · eLife Assessment]

The authors report that chemogenetic methods targeting the ventral cervical spinal cord can be used to increase phrenic inspiratory motor output and subsequent diaphragm EMG activity and ventilation in rodents. These findings are **important** because they are a necessary first step towards using chemogenetic methods to drive inspiratory activity in disorders in which motor neurons are compromised, such as spinal injury and degenerative disease. The data are **convincing**, with rigorous assessments of phrenic inspiratory activity and its ability to drive the diaphragm and subsequent ventilation, as well as assessments of DREADD expression.

---

## [Referee Report · Reviewer #1 (Public review)]

Summary:

In this manuscript, the authors report that activation of excitatory DREADDs in the mid-cervical spinal cord can increase inspiratory activity in mice and rats. This is an important first step toward an ultimate goal of using this, or similar, technology to drive breathing in disorders associated with decreased respiratory motor output, such as spinal injury or neurodegenerative disease. Strengths to this study include a comparison of non-specific DREADD expression in the mid-cervical spinal cord versus specific targeting to ChAT-positive neurons, and the measurement of multiple respiratory-related outcomes, including phrenic inspiratory output, diaphragm EMG activity and ventilation. The data show convincingly that DREADDs can be used to drive phrenic inspiratory activity, which in turn increases diaphragm EMG activity and ventilation.

Comments on revisions: All of my prior comments have been sufficiently addressed.

---

## [Referee Report · Reviewer #2 (Public review)]

Summary:

This study shows that when excitatory DREADD receptors are expressed in the ventral area of the cervical spinal cord containing phrenic motoneurons, systemic administration of the DREADD ligand J60 increases diaphragm EMG activity without altering respiratory rate. The authors took a non-selective expression approach in wild-type mice, as well as a more selective Cre-dependent approach in Chat-Cre mice and Chat-Cre rats to stimulate cervical motoneurons in the spinal cord. This is a proof of principle study that supports the use of DREADD technology to stimulate the motor output to the diaphragm.

Strengths:

The strengths of the study lie in the use of both mice and rats to test whether the chomogenetic activation of phrenic motoneurons with multiple experimental approaches increases diaphragm EMG activity (both tonic and phasic) and tidal volume.

Comments on revisions:

Thanks for addressing my comments. One last comment that could be discussed or addressed is :

Line 295- was the time post-infection, which varies considerably between groups and across samples, taken into consideration when comparison of response was made between ChatCre mice (4-9 weeks post-infection) and WT mice (four to five weeks post-infection)?

---

## [Author Response]

The following is the authors’ response to the current reviews.

**Reviewer #2:**
Line 295 – was the time post-infection, which varies considerably between groups and across samples, taken into consideration when comparison of response was between ChatCre mice (4-9 weeks post-infection) and WT mice (four to five weeks post-infection)?

Thank you for your comment. We did not originally assess the effects of time post-injection on DREADD response. Generally, AAV transgene expression has been demonstrated to be long-term and stable in the CNS of mice.[1] However, there is some variation in the reporting time of peak transgene expression[2], and this may potentially impact our results.

In investigating this issue further, we discovered an error in our reporting as we did have n = 1 wild-type mouse that underwent EMG recordings 62 days (~9 weeks) post-AAV injection. This has been corrected in the manuscript (lines 87-88).

Addressing this question is challenging due to the uneven distribution of time points within the 4–9-week windows for each group. Essentially, there were two groups per cohort, one studied at 4-5 weeks and one at 8-9 weeks. More specifically:

- Wild-type cohort: n = 10 animals were studied 28–33 days post-injection, and n = 1 at 62 days.

- ChAT-Cre cohort: n = 4 animals were studied 28–30 days post-injection, and n = 5 at 56–59 days.

We performed Pearson correlation analyses between time post-injection and diaphragm EMG response to DREADD activation (peak amplitude and area under the curve, AUC) for both cohorts (Author response image 1):

- ChAT-Cre: No significant correlations were found (peak amplitude: r^2^ = -0.117, r = -0.1492, p = 0.702, Figure 1a-b; AUC:r^2^ = -0.0883, r = 0.2184, p = 0.572, Figure 1c-d).

- Wild type: Initial analysis of all data showed significant correlations (peak amplitude:r^2^ = 0.362, r = 0.6523, p = 0.0296, Figure 1a; AUC: r^2^ = 0.347, r = 0.6424, p = 0.033, Figure 1c), suggesting a moderate positive correlation between time post-injection and EMG response. However, when the single 8–9-week wild-type mouse was excluded, these correlations were no longer significant (peak amplitude: r^2^ = 0.172, r = 0.5142, p = 0.128, Figure 1b; AUC: r^2^ = 0.23, r = 0.5614, p = 0.0913, Figure1d).

Comparing wild-type and ChAT-Cre groups directly was unreliable due to the single wild-type mouse studied at the later time point. We attempted to model time post-injection as a continuous variable (i.e., exact days post-injection) using a restricted maximum likelihood mixed linear model in JMP; however, the analysis could not be performed because there were not sufficient overlapping time points between the two cohorts (i.e., not all days post-injection were represented in both groups). To mitigate this, we binned animals into two groups: 4–5 weeks and 8–9 weeks post-injection. This analysis returned a significant interaction between cohort and time post-injection (p = 0.0391), however there were no significant multiple comparisons upon Tukey post hoc test (i.e., p > 0.05).

Based on these findings, we feel confident that time post-injection is unlikely to have a significant impact on diaphragm EMG response to DREADD activation in the ChAT-Cre cohort. However, in the wild-type cohort, it is difficult to draw definitive conclusions, as only one animal was studied at the 8–9-week time point. For similar reasons, it remains unclear whether the relationship between time post-AAV transduction and DREADD response differs between cohorts. Given the inconclusive nature of these results, we have elected not to include this analysis in the manuscript. Nevertheless, to ensure transparency, we have provided Author response image 1 below of peak amplitude and AUC plotted against time, allowing readers to evaluate the data independently.

**Author response image 1. sa3fig1:** Plots of diaphragm EMG peak amplitude, (a-b) and area under the curve (c-d) vs. days post-AAV injection for wild-type (blue) and ChAT-Cre (orange) mice. Pearson correlation analyses were performed to assess the relationship between time post-AAV injection and diaphragm EMG DREADD response in wild-type and ChAT-Cre mouse cohorts. r^2^, r, and p-values are shown in each panel for both cohorts. Panels a and c display peak amplitude and AUC, respectively, including all animals. Panels b and d present the same variables with the n = 1 wild-type mouse at the 9-week time point excluded; ChAT-Cre data is unchanged between corresponding panels. Scatter points represent data from individual animals. Polynomial trendlines are displayed for each cohort with wild-type in blue and ChAT-Cre in orange.

REFERENCES

(1) Kim, J. Y., Grunke, S. D., Levites, Y., Golde, T. E. & Jankowsky, J. L. Intracerebroventricular viral injection of the neonatal mouse brain for persistent and widespread neuronal transduction. J Vis Exp, 51863 (2014). https://doi.org/10.3791/51863

(2) Hollidge, B. S. et al. Kinetics and durability of transgene expression after intrastriatal injection of AAV9 vectors. Front Neurol 13, 1051559 (2022). https://doi.org/10.3389/fneur.2022.1051559

The following is the authors’ response to the original reviews.

**Response to reviewer’s public reviews:**

We chose the dose of J60 based on a prior publication that established that off-target effects were possible at relatively high doses[1]. The dose that we used (0.1 mg/kg) was 30-fold less than the dose that was reported in that paper to potentially have off-target responses (3 mg/kg). Further, Author response image 1 shows the results of experiments in which J60 was given to animals that did not have the excitatory DREADD expressed in the spinal cord. This includes a sample of mice (n = 2) and rats (n = 3), recorded from using the same diaphragm EMG procedure described in the manuscript. The figure shows that there was no consistent response to the J60 at 0.1 mg/kg in the “control experiment” in which the DREADD was not expressed in the spinal cord.

**Author response image 2. sa3fig2:** Diaphragm EMG response to J60 administrated to naïve rats and mice. Panel a-b show raw EMG values at baseline, following vehicle (saline) and J60 administration for the left and right hemidiaphragm. Panel c-d shows EMG values normalized to baseline. Neither One-way RM ANOVA (panel a-b) nor paired t-test (panel c-d) returned significant p values (p < 0.05).

**Response to specific reviewer comments:**

**Reviewer #1:**
How old were the animals at the time of AAV injection, and in subsequent experiments?

The wildtype cohort of mice were 7-9 weeks old at time of AAV injection and DREADD experiments took place 4-5 weeks after AAV injection. ChAT-Cre mice were 6-10 weeks old at time of AAV injection and DREADD experiments took place 4-9 weeks after AAV injection. ChAT-Cre rats were 2-5 months old at time of AAV spinal injection. These animals underwent plethysmography recordings 3-4 months post-AAV injection and subsequently phrenic nerve recording 3-8 weeks later. These details have been added to the Method section.

How many mice were excluded from electrophysiology experiments due to deteriorating electrode contact?

No mice were excluded from electrophysiology experiments due to deteriorating electrode contact. If you are referring to the n = 1 excluded ChAT-Cre mouse (line 368) this animal was excluded because it showed no histological evidence of DREADD expression (lines 200-206).

What was the urethane dose?

The urethane dose for phrenic nerve recordings was 2.1 g/kg. See methods section line 395.

A graphical timeline of the experimental progression for plethysmography and electrophysiology studies would enhance clarity.

A graphical timeline has been added. See Figure S6.

Significance indicators in the figures would greatly enhance clarity. It is a little awkward to have to refer to supplemental tables to figure out statistical differences.

Significance indicators have been added. See Figures 1, 2, 4, and 5

In Figures 1, 2, and 5, individual data points should be shown, as in Fig 4.

Thank you for this suggestion. We agree that, in general, it is best practice to scatter individual data points. However, when we drafted the new figures, it was apparent that including individual scatter points, in this case, created very “cluttered” figures that were very difficult to interpret.

More detail regarding the plethysmography studies is needed. Was saline/J60 infused via a tail vein catheter? Were animals handled during the infusion? How long is the "IV" period? What volume of fluid was delivered?

All IV infusions were delivered via a tail vein catheter. Animals were not handled during infusion nor at any point during the recording. An IV catheter was externalized via a port in the plethysmograph allowing for IV infusion without handling of the animal or opening the plethysmograph. The infusion period for both saline and J60 was standardized to 2 minutes. The volume of fluid of both saline and J60 was standardized to 0.6 mL. This information has been added to the methods section (lines 408-410, 415-16, 419-420).

**Reviewer #2:**
The abstract could be improved by briefly highlighting the rationale, scope, and novelty of the study - the intro does a great job of highlighting the scope of the study and the research questions.

A brief explanation of the rationale, scope, and novelty of the study has been added to the abstract. See lines 2-8.

Line 18, specifies that this was done under urethane anesthesia.

This detail has been added to the abstract (line 20).

The methods section should be moved to the end of the manuscript according to Journal policy.

The methods section has been moved to the end of the manuscript.

The authors mention the use of both female and male rats but it is not indicated if they tested for and observed any differences between sexes across experiments.

We included the use of both male and female animals in this study to improve the generalizability of the results. However, we were not adequately powered for sex comparisons and therefore did not perform any statistical analysis to assess differences between sexes across experiments. Text has been added to the methods section (lines 534-537) to clarify.

Line 40, since delivery of J60 was performed in both IV and IP, this general statement should be updated.

This detail has been revised to include both IV and IP. See line 43.

Line 42. "First, we determined if effective diaphragm activation requires focal DREADD expression targeting phrenic motor neurons, or if non-specific expression in the immediate vicinity of the phrenic motor nucleus would be sufficient...." I don't think that in the experiments with wild-type mice the authors can claim that they selectively targeted the cervical propriospinal network (in isolation from the motoneurons). Given the fact that the histological analysis did not quantify interneurons or motoneurons in the spinal cord, authors should be cautious in proposing which neuronal population is activated in the non-specific approach.

We agree, and this was a poorly worded statement in our original text. We agree that wild-type DREADD expression was not limited to the cervical propriospinal networks but likely a mix of interneurons and motoneurons. The text has been edited to reflect that (see lines 56-60).

AAV virus source is not described.

All AAVs were obtained from the UF Powell Gene Therapy Center. Details of virus source and production have been added to the methods section. See lines 336-347.

Line 108-125. Because the diaphragm EMG recordings are only described for mice here, I would suggest editing this methods section to clearly state mice instead of vaguely describing "animals" in the procedure.

“Animals” has been changed to “mice” to avoid ambiguity.

Line 120, add parenthesis.

Parenthesis has been added.

Line 126. Whole body plethysmography protocol. Three hypercapnic hypoxic challenges are a lot for a rat within a 3-hour recording session in freely behaving rats. Did the authors verify with control/ vehicle experiments that repeated challenges in the absence of J60 do not cause potentiation of the response? I understand that it is not possible to invert the order of the injections (due to likely long-term effects of J60) or it is too late to perform vehicle and J60 injections on different days, but controls for repeated challenges should be performed in this type of experiment, especially considering the great variability in the response observed in Figure 4 (in normoxic conditions).

We did not conduct control experiments to assess the impact of repeated hypercapnic hypoxic challenges on the naïve response (i.e., in the absence of J60). However, our experimental protocol was designed such that each experimental period (i.e., post-vehicle or post-J60 infusion) was normalized to baseline recordings taken immediately prior to the vehicle or J60 infusion. While repeated exposure to hypercapnic hypoxic challenges may have altered respiratory output, we are confident that normalizing each experimental period to its respective baseline effectively captures the impact of DREADD activation on ventilation, independent of any potential potentiation that may have occurred due to gas challenge exposure. We have included raw values for all plethysmography outcomes (see Figure 4, panels a-c) to ensure full data transparency. Still, we believe that the baseline-normalized values more accurately reflect the impact of DREADD activation on the components of ventilation.

Furthermore, why the response to the hypercapnic hypoxic challenges are not reported? These could be very interesting to determine the effects of DREADD stimulation on chemosensory responses and enhance the significance of the study.

Response to the hypercapnic hypoxic challenges has been added to the manuscript. See Figure S3 and results section lines 162-167. Briefly, there were no statistically significant (p < 0.05) differences in tidal volume, respiratory rate, or minute ventilation between J60 vs sham condition during hypercapnic-hypoxic ventilatory challenges.

Line 200 - what is the reason behind performing a qualitative analysis of mCherry in various quadrants? This limits the interpretation of the results. If the authors used Chat-cre rats, the virus should only be in Chat+ MN. Knowing how selective the virus is, and whether its expression was selective for Phrenic MN versus other MN pools, could address several technical questions.

We agree that detailed quantification of expression by motoneuron pool would be of value in future work. However, for these initial proof-of-concept experiments, we performed the quadrant-based qualitative analysis of mCherry expression to provide a simple comparison of mCherry expression between groups (i.e., ChAT-Cre vs. wildtype mice). This analysis allowed us to: (1) show the reader that each animal included in the study showed evidence of mCherry expression and (2) give the reader an idea of patterns of mCherry expression throughout the mid-cervical spinal cord. Additionally, it is important to note that while ChAT is a marker of motoneurons some populations of interneurons also express ChAT(2-4).

Given the increased values of Dia EMG AUC and no changes in respiratory rate, did the authors determine if there was a change in the inspiratory time with J60 administration?

We did not assess inspiratory time.

High death rate in DREADD WT mice - was histological analysis performed on these mice? Could it be due to the large volume injected into the spinal cord that affects not only descending pathways but also ascending ones? Or caused by neuronal death due to the large volume of viral solution in injected in mice.

Histological analysis was performed on these animals to assess mCherry expression only (i.e., no staining for NeuN or other markers was performed). While the reviewer's speculations are reasonable, we feel these reasons are unlikely to explain the death rate in DREADD WT mice as ChAT-Cre mice received the same volume injected into their spine and lived up until and during diaphragm EMG recordings. Additionally, WT mice lived for 4-5 weeks post-injection which would be past the acute phase that a large immune response to the viral dose would have occurred.

Line 299-304. Can you please clarify whether these rats were tested under anesthesia?

These rats were assessed under anesthesia. This detail has been added (line 146).

Given some of the unexpected results on cardiovascular parameters in urethane anesthetized rats, did the authors test the effects of J60 in the absence of AAV construct infection?

A small cohort (n = 2) of urethane anesthetized naïve wildtype rats were given the J60 ligand (IV, 0.1 mg/kg dose). We did observe a sudden drop in blood pressure after J60 administration that was sustained for the duration of the recording. One animal showed a 12% decrease in mean arterial blood pressure following J60 administration while the other showed a 35% decrease. Thus, it does appear that in this preparation the J60 ligand is producing a drop in arterial blood pressure.

Line 393. I believe this comment is referred to the intrapleural and diaphragmatic injection. Maybe this should clarified in the sentence.

This sentence has been revised for clarity (see lines 248-250).

Figures 1 and 2. It would be informative to show raw traces of the Diaphragm EMG to demonstrate the increase in tonic EMG. It is not possible to determine that from the integrated traces in Figures 1A and B.

Thank you for bringing up this concern. While the mean data in Figures 1F and 2F do indicate that, on average, animals had tonic diaphragm EMG responses to DREADD activation, the examples given in Figures 1A and 2A show minimal responses. This makes it difficult to fully appreciate the tonic response from those particular traces. However, clear tonic activity can be appreciated from Figures 5A and S2. In these figures, tonic activity is evident from the integrated EMG signals, presenting as a sustained increase in baseline activity between bursts—essentially an upward shift from the zero point.

References

(1) Van Savage, J. & Avegno, E. M. High dose administration of DREADD agonist JHU37160 produces increases in anxiety-like behavior in male rats. *Behav Brain Res* 452, 114553 (2023). https://doi.org/10.1016/j.bbr.2023.114553

(2) Mesnage, B. *et al.* Morphological and functional characterization of cholinergic interneurons in the dorsal horn of the mouse spinal cord. *J Comp Neurol* 519, 3139-3158 (2011). https://doi.org/10.1002/cne.22668

(3) Gotts, J., Atkinson, L., Yanagawa, Y., Deuchars, J. & Deuchars, S. A. Co-expression of GAD67 and choline acetyltransferase in neurons in the mouse spinal cord: A focus on lamina X. *Brain Res* 1646, 570-579 (2016). https://doi.org/10.1016/j.brainres.2016.07.001

(4) Alkaslasi, M. R. *et al.* Single nucleus RNA-sequencing defines unexpected diversity of cholinergic neuron types in the adult mouse spinal cord. *Nat Commun* 12, 2471 (2021). https://doi.org/10.1038/s41467-021-22691-2